# The relative impact of interventions on sympatric *Plasmodium vivax* and *Plasmodium falciparum* malaria: A systematic review

**Melanie Loeffel**[1,2], **Amanda Ross**[1,2]*

**1** Swiss Tropical and Public Health Institute, Allschwil, Switzerland, **2** University of Basel, Basel, Switzerland

\* amanda.ross@unibas.ch

**Data Availability Statement:** The data are in the Supporting Information files and at https://github.com/rossaman4/vivax-falciparum-relative-impact.

## Abstract

### Background

In areas with both *Plasmodium vivax* and *Plasmodium falciparum* malaria, interventions can reduce the burden of both species but the impact may vary due to their different biology. Knowing the expected relative impact on the two species over time for vector- and drug-based interventions, and the factors affecting this, could help plan and evaluate intervention strategies.

### Methods

For three interventions (treated bed nets (ITN), mass drug administration (MDA) and indoor residual spraying (IRS)), we identified studies providing information on the proportion of clinical illness and patent infections attributed to *P. vivax* over time using a literature search. The change in the proportion of malaria attributed to *P. vivax* up to two years since implementation was estimated using logistic regression accounting for clustering with random effects. Potential factors (intervention type, coverage, relapse pattern, transmission intensity, seasonality, initial proportion of *P. vivax* and round of intervention) were assessed.

### Results

In total there were 55 studies found that led to 72 series of time-points for clinical case data and 69 series for patent infection data. The main reason of study exclusion was insufficient information on interventions. There was considerable variation in the proportion of malaria attributed to *P. vivax* over time by study and location for all of the interventions. Overall, there was an increase apart from MDA in the short-term. The potential factors could not be ruled in or out. Although not consistently significant, coverage, transmission intensity and relapse pattern are possible factors that explain some of the variation found.

### Conclusion

While there are reports of an increase in the proportion of malaria due to *P. vivax* following interventions in the long-term, there was substantial variation for the shorter time-scales considered in this study (up to 24 months for IRS and ITN, and up to six months for MDA).

**Funding:** The authors received no specific funding for this work.

**Competing interests:** The authors have declared that no competing interests exist.

The large variability points to the need for the monitoring of both species after an intervention. Studies should report intervention timing and characteristics to allow inclusion in systematic reviews.

## Author summary

In areas with both *P. vivax* and *P. falciparum* malaria, an intervention may impact the species differently due to their different biology. In particular, an inoculation with *P. vivax* can lead to liver-stage parasites which relapse weeks or months later. The intervention may affect the species equally or one more than the other and this may change over time. Knowledge of the expected relative effect of interventions would be useful for tailoring intervention strategies to address both species, and for interpreting monitoring data. This study aims to assess patterns of the proportion of clinical cases or patent infections that were *P. vivax* or *P. falciparum* on a short time scale of up to two years since implementation and to identify potential factors that could explain variation in these observed patterns for two vector-based interventions (insecticide treated nets and indoor residual spraying) and a drug-based intervention (mass drug administration). In all, there were 72 series of time-points for clinical data and 69 for patent infections. Overall there was an increasing proportion of *P. vivax* over time, but there was substantial variation between studies for all interventions. Coverage, transmission and relapse pattern were identified as possible explanations for some of the variation, however, this remains uncertain due to the high variation and the data available.

## Introduction

Of the species of malaria infecting humans, *Plasmodium falciparum* and *Plasmodium vivax* occur the most frequently and together accounted for 241 million cases in 2020 [1]. Separately they have different geographical ranges, but they co-exist particularly in the horn of Africa and Madagascar, South and Central America, the South-East Asian region and the Western Pacific region [2,3].

Both vector- and drug-based interventions can affect both species. In areas where both species are prevalent, the impact of the interventions may differ. Different patterns of the ratio of *P. vivax* to *P. falciparum* cases over time have been reported. One study in Thailand [4] with treatment of cases reported equal reductions in both species. Another in Papua New Guinea [5], where the usage of nets was increased found that *P. vivax* became dominant. A recent review by Price [6] found that control often leads to an increase in the ratio of malaria attributed to *P. vivax* over longer periods and describes studies where initial effects are a decrease in *P. falciparum* malaria followed by a decrease in *P. vivax* later [6–8] suggesting different dynamics over time.

Possible explanations for different relative effects can lie in biological differences, particularly the ability of *P. vivax* to form dormant hypnozoites in the liver which can relapse weeks or months later [9]. The relapses are likely to decrease the time to patent re-infection following clearance of blood-stage parasites by drugs, and to lower the impact of vector control on reducing blood-stage infections until the relapses have been depleted. Other possible factors are characteristics of the setting such as transmission intensity or seasonality, and characteristics of the intervention, such as the mode of action, coverage, and whether it is the first

implementation or a continuation of previous implementations since the dynamics for these can differ [10].

Program managers and decision-makers need to know the likely impact of interventions in their settings. Knowledge of the expected relative impact over time and the drivers in different settings may also allow easier interpretation of data for evaluating the impact of an intervention strategy, allow the tailoring of combinations of interventions for the optimal impact on both species and inform epidemiology.

We investigate the relative impact of vector-control and drug-based interventions on *P. vivax* and *P. falciparum* on a short-term time scale up to two years after implementation. We carry out a literature search to identify studies in areas with both *P. vivax* and *P. falciparum* which have documented distributions of treated bed nets (ITN), indoor residual spraying (IRS) or mass drug administration (MDA). We then estimate the variation in the observed trends of the proportions of clinical cases and patent infections that are attributed to *P. vivax* after the implementation of the interventions. We also aim to identify factors that affect the change in the proportion of clinical cases or patent infections.

## Methods

### Literature search

**Identifying potential studies.**  The search terms for the PubMed search were: (("falciparum"[tiab] OR "Malaria, Falciparum"[Mesh] OR "Plasmodium, Falciparum"[Mesh]) AND ("vivax"[tiab] OR "Malaria, Vivax"[Mesh] OR "Plasmodium, Vivax"[Mesh])) AND ("Insecticides/administration and dosage"[Mesh] OR "Mass Drug Administration"[Mesh] OR "Insecticide-Treated Bednets"[Mesh] OR "IRS"[tiab] OR "indoor residual spraying"[tiab] OR "Prevalence"[Mesh] OR "Incidence"[Mesh]). This search in PubMed generated 973 hits (03.08.2020). Published reviews were also screened. Cochrane reviews on mass drug administration (MDA) [11], indoor residual spraying (IRS) [12], insecticide treated nets (ITNs) [13,14] and IRS and ITN [15] were screened for studies that took place in areas with both malaria species. Furthermore, if papers that were found with the other two approaches cited papers that could potentially be useful, then these were included for screening as well.

Papers identified as potentially relevant were imported into Endnote for abstract screening. Studies that were not in English or German were excluded. Studies that based on the abstract did not fulfill the inclusion criteria (Box 1) were not considered potentially relevant. If relevance was unclear, they would be included in the full-text screening. In particular, the interventions implemented were frequently unclear based on the abstract alone.

---

Box 1

Inclusion criteria

1. Report both *P. vivax* and *P. falciparum* in the same area

2. Report incidence and/or prevalence on at least two time-points or have a concurrent control group

3. Intervention in between two surveys or control groups that did not receive the intervention

---

### Full-text screening of papers identified as potentially relevant

The papers considered as potentially relevant based on the abstract underwent full-text screening to see if they fulfilled the inclusion criteria (Box 1). This was carried out by one reviewer (ML) and discussed in detail with a second reviewer (AR).

Some studies would fulfill these criteria, however, needed to be excluded due to various reasons (Box 2).

---

Box 2

Exclusion criteria

1. Insufficient information about the intervention

2. Only one time-point measured

3. Epidemic malaria

4. Cases of one species being mainly due to imported cases

5. Population consisting solely of pregnant women

---

The information on the timing and coverage of interventions needed to be sufficiently detailed. For all three interventions, in most cases the starting point could be pinpointed to the month. For IRS and MDA the starting point needed to be within a window of not more than two years but the interventions could happen over longer time periods. In case of treated bed nets there needed to be a distribution (not just the availability of treated bed nets) with the time-point of the distribution within a window of two years.

Some studies would report having malaria control programs but did not indicate information on timing or control interventions and therefore were excluded. In other cases, there were different interventions implemented at the same time which would not allow disentangling the impacts and therefore were excluded. Interventions were included no matter if it was the first time they happened or a repeated time, but this was noted in order to be able to distinguish between first or repeated rounds of intervention.

Further interventions such as repellents, prophylaxis and larviciding were beyond the scope of this review and were excluded. However, studies were included if they did such interventions as well as those of interest and a note was taken of the additional intervention. Randomized control studies could be included if either they reported measurements before the intervention was implemented as well or if they provided a control group.

### Data extracted from the papers and additional variables from other sources

Extracted information was transferred into a Microsoft Access database. Several manuscripts could be combined if they reported on the same area to give one series of time-points. Similarly, if one study reported different areas or different age groups this would be added into the database separately leading to several series of time-points from one study.

## Outcome variables

The number of clinical cases and prevalence of patent infections were extracted from the studies. If the studies did not include incidence or prevalence as a table or in the text but had a figure the web plot digitizer [16] was used to extract the values of the data points.

## Rationale for selection of variables

**Intervention and round of intervention.**    Drug- and vector-based interventions have different effects over time. For MDA, the initial impact occurs quickly by clearing blood-stage infections. However, MDA does not protect against new blood-stage infections beyond the prophylactic period of the drugs [17]. Vector-based interventions do not clear existing blood-stage infections but reduce the number of new infections from infected bites. For both vector- and drug-based interventions, new blood-stage infections can come from infected bites for both species and additionally, unless the hypnozoites are cleared or used up, *P. vivax* relapses. As a consequence, the proportion of cases attributed to *P. vivax* may increase in the short-term.

A modelling study on *P. falciparum* found that when nets are implemented for the first time, a drop in transmission is expected, whereas repeated rounds are expected to sustain the previous reduction or led to an increase in transmission via changes in the levels of acquired immunity in the community [10]. Therefore, the dynamics can differ according to round of intervention.

**Coverage.**    If an intervention does affect one species more than the other, we would expect increased coverage to lead to a stronger change in the proportion.

**Transmission intensity.**    Modelling studies have predicted stronger effects of ITN [10,18] and MDA [19] for *P. falciparum* in low transmission settings. The transmission intensities of *P. falciparum* and *P. vivax* are frequently different in the same setting.

**Age.**    The age at the peak clinical incidence can differ between species in a setting, due both to their transmission intensities and the rate at which acquired immunity is gained for the species. As a consequence, interventions may lead to different impacts for the species.

**Relapse pattern.**    The number and timing of the relapses per infected bite are likely to influence the timing of any changes in the proportion of cases that are *P. vivax*.

**Seasonality.**    The effect of an intervention may vary depending on how seasonal the setting is, and the degree of seasonality can differ by species.

**Season when the data was collected.**    The proportion of *P. vivax* to *P. falciparum* differs by the season that the measurement was taken in [20–22] due to the ability of *P. vivax* to relapse.

## Definitions of variables

**Intervention and round of intervention.**    The round of intervention was noted for MDA and ITN. The spraying rounds of IRS were not added individually but the starting point was noted. It was often not possible to distinguish between first and later rounds for IRS, since spraying in the previous year was not reported. Spraying was classed as repeated if spraying the year before was reported. In this case, the studies usually looked at a change in the insecticide used.

For ITN distribution, we assumed that if no previous distribution was mentioned then none had happened. An exception to this was a study by Deressa [23] where 65% of people already owned nets. Therefore, it was assumed there had been a previous distribution although not specifically mentioned. There was no distinction made between insecticide treated nets (ITNs) and long lasting insecticidal nets (LLINs).

Studies were excluded when the previous intervention was likely to still have an effect: interventions were considered to not influence the effect of the new intervention substantially if the previous intervention was started more than six months before the intervention of interest, if the concurrent intervention was an increase in case management, or if the study gave reasons to believe that the previous intervention was not very impactful.

**Coverage.**   Studies were categorized into higher or lower coverage. An arbitrary cut-off of 70% was used for ITNs based on an equal division of the studies.

The studies had various ways of quantifying ITN coverage. The preferred metric of coverage for bed nets was usage followed by the metrics listed in Box 3. In many cases only one of these measures was available.

Box 3

Quantifying the Coverage of ITN

1. Usage of bed nets

2. Coverage of bed nets

3. Calculation using number of nets and population assuming 1 net covers 2 people

4. If usage in different seasons is given, it was assumed to be high if majority of seasons had usage above 70%

5. If self-reported usage was given for several years after distribution, first self-reported usage was considered

6. With several locations but coverage information for one only, it was assumed to be the same for other locations too.

For one study (Deressa [23]) the aim was to cover with one net per two people. It was not clear if this was reached but assumed it was and therefore the coverage would be high. In other cases, there were studies (Luxemburger [24], Rowland [25], Rowland [26]) where only a fraction of the community would receive nets and only those people would be tested. However, as only the people who received nets were tested the coverage was defined on what proportion of these people received a net. We recognize that community effects of the nets would be reduced in this situation.

As MDA and IRS generally had higher coverages, the cut off was set to 90% (of people receiving a dose or of households being sprayed). If there were several rounds of MDA coverage was considered high if 90% or more received at least one round.

**The number of months since the intervention was implemented.**   The studies usually gave a time-span of the implementation of the intervention, which could be the year of implementation but was usually more exact dates. To define when the intervention was implemented (time zero), the mid-point of these time spans was taken. Similarly, the mid-point of time windows given for prevalence surveys and incidence data collection were used. In most series of MDA, there were several rounds over three months. The mid-point of the time span of the first round was used as the time of implementation.

The proportion of *P. vivax* cases just before the intervention was implemented is needed to allow the change in proportion over time to be estimated. To determine the proportion at this time-point, the closest time-point before the intervention was used unless there were several data points within six months before the intervention, in which case all of these were used in order to to provide a more stable estimate of the proportion.

However, some studies had measured malaria burdens more than six months before the intervention, these were included using the closest measurement before the intervention was used to avoid excluding many studies (for clinical case data: 15 series of time points (21%) (median time point of measurement of these series: 15 months before intervention), for patent infection data: 32 series of time points (46%) (median time point of measurement of these series: 30 months before intervention)). We assumed that if there had been no major interventions, then the proportion would be similar after accounting for season of measurement in the regression analyses. The median time of the pre-implementation points was 0.2 months for clinical cases and 9.7 months for patent infections. The data-points after the intervention was implemented were used up to the time of another implementation. This means that a data-point could be used twice, once as a measurement after an intervention happened and once before a different intervention. In all the analyses, the season of the measurement was accounted for.

**Transmission.**    The transmission intensities of *P. falciparum* and *P. vivax* frequently differ in a setting. *P. vivax* and *P. falciparum* were categorized separately into high and low transmission intensities which led to four transmission categories. The transmission intensity in the study settings was determined using malaria burden measures from the studies, however, the studies had non-identical measures of the incidence of clinical cases and prevalence. For example, incidence data could be reported in different age groups or collected actively or only passively. However as only a rough distinction between high and low transmission intensity was required, the information in the studies was used. If there were entomological inoculation rates given or prevalence these were used. An arbitrary cut off of 5% prevalence of *P. falciparum* and *P. vivax* was used to allow reasonable numbers of studies in the transmission categories.

When prevalence data was not available then incidence data was used. A transmission setting was considered high with more than 100 cases per 1000 people per year. In uncertain cases, other literature from the setting or maps from the Malaria Atlas Project would be consulted [2,3]. However, these maps are recent and only go back to 2000 while some of the studies were carried out in previous decades and therefore these maps could not be used for all studies.

**Relapse pattern.**    To define the relapse pattern of *P. vivax* occurring in the study areas, the classification by White [27] was used. White distinguishes between long latency relapse patterns (relapses approximately after 8–10 months) and frequent relapse patterns with relapses shortly after the primary infection. The classification is based on the broad geographical regions in Fig 17 in White [27]: regions may have frequent relapse strains only, long latency strains only or both. Latency is also associated with the number of relapses with temperate strains displaying fewer relapses than tropical strains [27].

**Seasonality.**    Seasonality of rainfall was used as a proxy for the seasonality of malaria transmission [28]. Although this is not the only factor that influences seasonality, it was readily available for all areas. The relative entropy quantifies how much the distribution of rainfall differs from a uniform distribution of the rain across all months of the year [29]. To assign seasonality, the model based on the Climate Prediction Center Merged Analysis of Precipitation dataset (CMAP model) of the relative entropy [29] was used. The threshold chosen to distinguish between low and high seasonality was an arbitrary value of 0.6.

**Seasonal patterns.**    The season at the time-point of measurement was often reported in the studies. When it was missing other studies with information on the timing of seasons in the area were used. If the prevalence or incidence was an average over the year, the season of survey was assumed to be in both wet and dry seasons. If the season the data was collected in could not be identified, it was noted as missing.

**Sample size of studies.**    The actual case numbers or number of people infected were extracted.

**Proportion of malaria attributed to P. vivax.**    To calculate the proportion of malaria cases with *P. vivax* infection, the number of clinical cases (or patent infections) with *P. vivax* infection was divided by the number of clinical cases (or patent infections) with either *P. vivax* or *P. falciparum*. If the studies counted mixed cases separately, they were added to both *P. vivax* case numbers as well as *P. falciparum* case numbers, as this was mostly reported to have been done in the studies when the mixed case numbers were not given separately. There were studies where mixed cases counted towards one species only, this was noted in Table A-J in S1 Supporting Information. Only malaria cases with *P. vivax* and/or *P. falciparum* were used in the analysis.

## Risk of bias

The risk of bias was assessed in general rather than for each individual study. None of the studies aimed to assess the relative impact of an intervention on *P. vivax* and *P. falciparum* directly, and so the details which may affect the risk of bias were generally not reported. We considered the seven domains of the ROBINS-I tool (confounding, selection of participants, bias in classification of an intervention, deviation from intended interventions, missing data, bias in measurement, selective reporting depending on findings) [30].

## Statistical analysis

The analysis aimed to (i) describe changes in the proportion of cases attributed to *P. vivax* by time since the implementation of the intervention and (ii) to identify factors that influence the pattern. Separate analyses were carried out for clinical cases and patent infections. R (version 4.0.1) [31] was used.

To display the data visually, the proportion attributed to *P. vivax* was plotted over time for each series individually (number of *P. vivax* cases divided by number of *P. vivax* and *P. falciparum* cases combined). The size of the point increased with higher number of cases found at that time-point. Spaghetti plots by factor gave a visual impression of the potential role of the factor.

Characteristics of the series of time-points, such as age-group, were summarized using proportions. R-packages used for data processing and the visualization of data were diagrammeR [32], maps [33], ggplot2 [34], rnaturalearth [35], plyr [36], reshape2 [37].

Logistic regression was used for the analysis with the proportion of either clinical cases or patent infections that were *P. vivax* as the outcome. The change in the proportion of *P. vivax* over time was described as the change in odds per month since implementation. The model includes random effects for both intercept and slope to account for clustering by series. Time was used as a continuous variable. Potential explanatory variables were included both as the variable alone and as an interaction with time, to assess their effect on the change in proportion over time.

Maximum likelihood was used to fit the models. The lme4 package [38] in R was used. In all models, the season of the survey was adjusted for since season is likely to affect the proportion of *P. vivax* to *P. falciparum*.

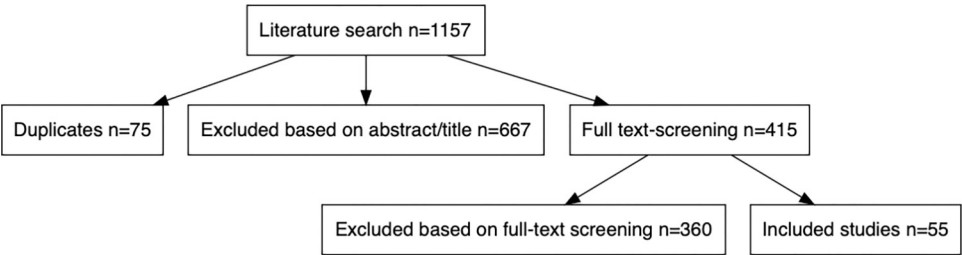

**Fig 1. Flow diagram of studies in the review.**

Randomized controlled studies with control groups were included. If no measurements before the intervention were available, it was assumed that before the intervention the proportion of *P. vivax* to *P. falciparum* was equal to the proportion in the control group that did not receive the intervention. This is equivalent to assuming the effect observed in the study would also have been observed longitudinally.

## Ethics statement

This study did not require any special ethical clearance. No new data were collected: the data included in the review were extracted from published papers.

## Results

### Literature search

The literature search lead to 1157 hits of which 55 studies were included (Fig 1). The PubMed search yielded 973 hits, and 184 were obtained by screening citations in Cochrane Reviews and the included papers.

The full-text screening included 415 studies. The most common reason for exclusion was the lack of information on interventions (Table 1), either no interventions were reported, or it was not clear at what time-point interventions happened or what the intervention was. In 56 cases there were studies that either only did a survey at one time-point or if there were surveys at several time-points the results would be an average of these surveys rather than reporting

**Table 1. Reasons for exclusion.**

| Reason for exclusion | Number of studies |
|---|---|
| Lacks information on intervention or no intervention | 208 |
| No distinction of different time-points of the surveys | 56 |
| No distinction of species across time | 52 |
| One species not detected or not reported | 13 |
| Imprecise because data is for the whole country | 7 |
| Same data as another study that is already used | 6 |
| Not outcome of interest | 6 |
| Looks at epidemics | 4 |
| Issues with disentangling intervention effect | 5 |
| Very low prevalence of *P. falciparum* | 1 |
| Excluded relapses of *vivax* from cases | 1 |
| Imported cases | 1 |
| **Total** | **360** |

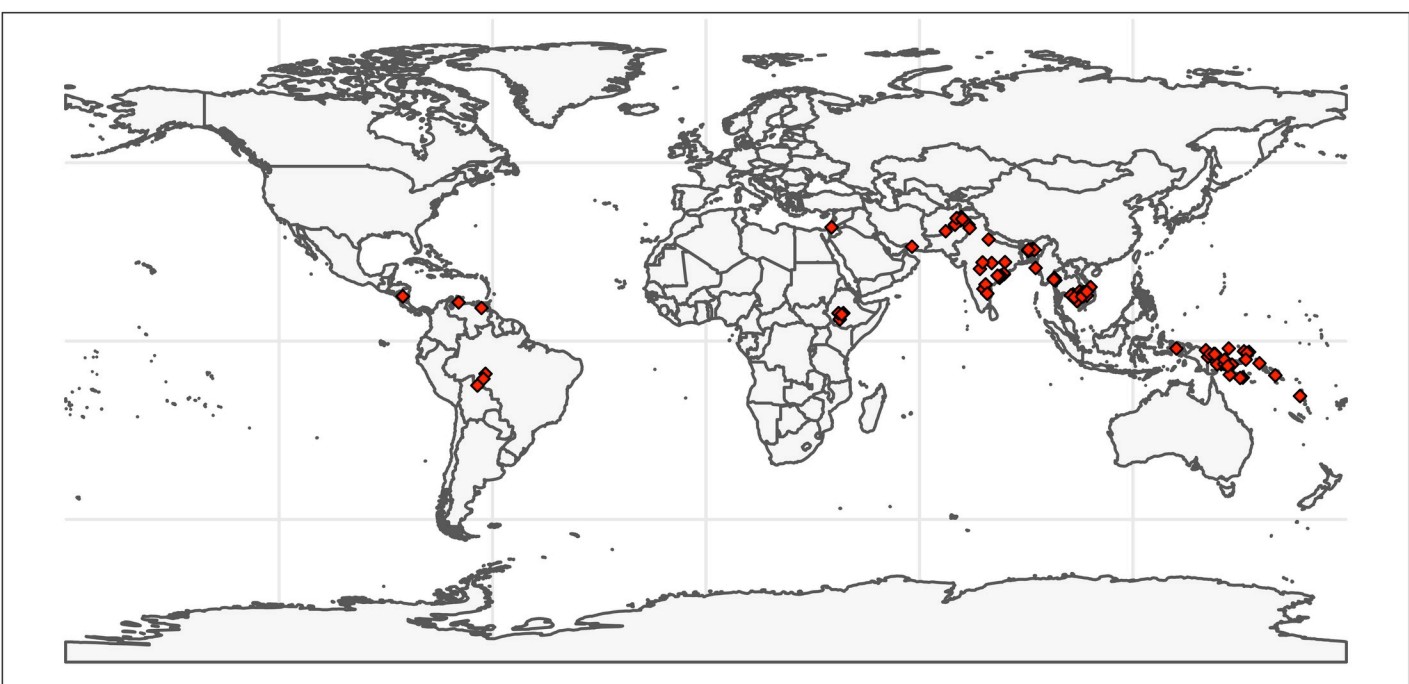

**Fig 2. Location of the studies.** Made with Natural Earth (https://www.naturalearthdata.com/ and downloaded using the R package rnaturalearth.

them separately. In a further five studies it was not possible to disentangle the effect of the interventions because there were several interventions happened at the same time.

A broad description of the included studies can be found in S1 Supporting Information. The largest number of studies was in Papua New Guinea (nine studies, 20%) followed by India (eight studies, 17%). The remaining studies were located in Nicaragua, Venezuela, Brazil, Ethiopia, Palestine, Iran, Pakistan, Afghanistan, Cambodia, Vietnam, Myanmar, Thailand, Indonesia, Vanuatu and the Solomon Islands (Fig 2).

Some studies reported several interventions (not at the same time), or several locations, and could therefore provide multiple series of time-points. Others were combined since they reported on the same area. In total, there were 141 series of time-points, 69 of patent infections and 72 of clinical cases. For three ITN distributions [39–41], the round of intervention was unclear or another intervention was phased out as the distribution took place and therefore they were excluded from the analysis.

## Summary of the data

Twenty-seven studies met the criteria to be included in the analyses of clinical cases, providing 72 series of time-points for different age-groups and locations and a total of 500 time-points. Overall, 16 (22%) of the series included IRS, 39 (54%) treated nets, and 17 (24%) drug treatment (Table 2). The study dates ranged from 1959 to 2019 with 21 series of time-points from studies before the year 2000 (range ITN: 1996–2019, range MDA: 1983–2019, range IRS: 1959–2019).

The majority of the series of time-points included all age-groups, with only two series focusing on under 5-year-olds and two series on under 10-year-olds (Table 2 and S1 Supporting Information). Most of the series (92%) were observational and compared the clinical cases before and after the intervention was implemented. For six series (8%) there were control

**Table 2. Characteristics of the 72 series of time-points of clinical cases by intervention.**

| | ITN (first distribution) (n = 26) | ITN (repeated distribution) (n = 13) | MDA (first round) (n = 13) | MDA (repeated round) (n = 4) | IRS (first spraying) (n = 4) | IRS (repeated spraying) (n = 3) | IRS (undetermined round) (n = 9) | Overall (n = 72) |
|---|---|---|---|---|---|---|---|---|
| **Age group** | | | | | | | | |
| 0–5 years | 1 (4%) | 1 (8%) | 0 (0%) | 0 (0%) | 0 (0%) | 0 (0%) | 0 (0%) | 2 (3%) |
| 0–10 years | 0 (0%) | 0 (0%) | 0 (0%) | 0 (0%) | 1 (25%) | 1 (33%) | 0 (0%) | 2 (3%) |
| 10+ years | 0 (0%) | 0 (0%) | 0 (0%) | 0 (0%) | 0 (0%) | 1 (33%) | 0 (0%) | 1 (1%) |
| all ages | 25 (96%) | 12 (92%) | 13 (100%) | 4 (100%) | 3 (75%) | 1 (33%) | 9 (100%) | 67 (93%) |
| **Coverage[1]** | | | | | | | | |
| low | 9 (35%) | 4 (31%) | 10 (77%) | 4 (100%) | 0 (0%) | 0 (0%) | 0 (0%) | 27 (38%) |
| high | 15 (58%) | 6 (46%) | 2 (15%) | 0 (0%) | 1 (25%) | 3 (100%) | 7 (78%) | 34 (47%) |
| missing | 2 (8%) | 3 (23%) | 1 (8%) | 0 (0%) | 3 (75%) | 0 (0%) | 2 (22%) | 11 (15%) |
| **Seasonality[2]** | | | | | | | | |
| low | 18 (69%) | 0 (0%) | 11 (85%) | 4 (100%) | 0 (0%) | 3 (100%) | 4 (44%) | 40 (56%) |
| high | 8 (31%) | 13 (100%) | 2 (15%) | 0 (0%) | 4 (100%) | 0 (0%) | 5 (56%) | 32 (44%) |
| **Relapse pattern** | | | | | | | | |
| frequent | 13 (50%) | 10 (77%) | 3 (23%) | 0 (0%) | 0 (0%) | 0 (0%) | 1 (11%) | 27 (38%) |
| long | 4 (15%) | 0 (0%) | 9 (69%) | 4 (100%) | 2 (50%) | 3 (100%) | 3 (33%) | 25 (35%) |
| both | 9 (35%) | 3 (23%) | 1 (8%) | 0 (0%) | 2 (50%) | 0 (0%) | 5 (56%) | 20 (28%) |
| **Transmission[3]** | | | | | | | | |
| low Pf low Pv | 15 (58%) | 9 (69%) | 9 (69%) | 4 (100%) | 3 (75%) | 1 (33%) | 5 (56%) | 46 (64%) |
| low Pf high Pv | 3 (12%) | 0 (0%) | 2 (15%) | 0 (0%) | 0 (0%) | 0 (0%) | 2 (22%) | 7 (10%) |
| high Pf low Pv | 6 (23%) | 3 (23%) | 0 (0%) | 0 (0%) | 1 (25%) | 2 (67%) | 0 (0%) | 12 (17%) |
| high Pf high Pv | 2 (8%) | 1 (8%) | 2 (15%) | 0 (0%) | 0 (0%) | 0 (0%) | 2 (22%) | 7 (10%) |
| **Diagnostic tool** | | | | | | | | |
| RDT | 1 (4%) | 0 (0%) | 1 (8%) | 0 (0%) | 1 (25%) | 0 (0%) | 0 (0%) | 3 (4%) |
| RDT or microscopy | 10 (38%) | 0 (0%) | 2 (15%) | 0 (0%) | 0 (0%) | 0 (0%) | 0 (0%) | 12 (17%) |
| RDT then microscopy | 0 (0%) | 7 (54%) | 0 (0%) | 0 (0%) | 0 (0%) | 0 (0%) | 0 (0%) | 7 (10%) |
| microscopy | 8 (31%) | 4 (31%) | 10 (77%) | 4 (100%) | 3 (75%) | 3 (100%) | 8 (89%) | 40 (56%) |
| not reported | 7 (27%) | 2 (15%) | 0 (0%) | 0 (0%) | 0 (0%) | 0 (0%) | 1 (11%) | 10 (14%) |
| **Type of study** | | | | | | | | |
| observational | 23 (88%) | 13 (100%) | 13 (100%) | 4 (100%) | 3 (75%) | 3 (100%) | 7 (78%) | 66 (92%) |
| trial | 3 (12%) | 0 (0%) | 0 (0%) | 0 (0%) | 1 (25%) | 0 (0%) | 2 (22%) | 6 (8%) |
| **Survey taken in different seasons[4]** | | | | | | | | |
| yes | 6 (23%) | 2 (15%) | 12 (92%) | 4 (100%) | 1 (25%) | 2 (67%) | 7 (78%) | 34 (47%) |
| no | 20 (77%) | 11 (85%) | 1 (8%) | 0 (0%) | 3 (75%) | 1 (33%) | 2 (22%) | 38 (53%) |
| **Time points per time series** | | | | | | | | |
| mean | 4.3 | 5.8 | 14 | 6.3 | 6.8 | 4.7 | 7.2 | 6.9 |
| median (min, max) | 5 (2,9) | 3 (2,24) | 12 (5,34) | 6 (6,8) | 3.5 (2,18) | 5 (4,5) | 4 (2,15) | 5 (2,34) |

[1]High coverage is defined as more than 70% of people with access to intervention for ITN and 90% of people for IRS. For MDA high coverage is defined as more than 90% receiving at least one treatment

[2]Seasonality is defined using the measure of relative entropy described by Pascale (2015) [29] which quantifies the distribution of rainfall across the year

[3]Transmission intensity cut off was at 5% prevalence for the respective species or less than 100 cases per 1000 people per year

[4]The data collection may spread across different seasons or all the data points described may be collected throughout the same season (e.g. always during the wet season)

groups as comparison instead of clinical cases before the intervention was implemented. In more than half of the series of time-points microscopy was used as the diagnostic tool. Most series were set in areas with low transmission for both *P. vivax* and *P. falciparum*. The total number of clinical cases found per time-point ranged from 1 to 19020 cases.

There were 24 studies included with information on the prevalence of patent infections (Table 3), leading to a total of 69 series of time-points and 198 individual time-points. There were 18 (26%) series of time-points where IRS was conducted, 41 (59%) had treated bed nets and 10 (14%) with MDA. The study dates ranged from 1931 to 2020 with 23 series of time-points from studies before 2000 (range ITN: 1993–2020, range MDA: 1931–2019, range IRS: 1961–2005).

The time-window of intervention implementation for clinical case data was mostly one month (55%), and never longer than a year. For prevalence, the implementation time-span was within a month for 63% of the series of time points, and for 98% within a year. The time-span for clinical case data collection was monthly data in 69% of the series of time points and for 97% of the studies it was less than a year. For patent infection data, the time-span of collection was a month or less for 65% of the series of time points and all but one had collection time windows of a year or less. [42]. The total number of patent infections found per time-point ranged from 1 to 789 cases.

## Risk of bias

The majority of studies are before-after studies where the intervention is the same for all participants included at the time of implementation and the same participants are followed-up for both *P. vivax* and *P. falciparum* infection. Consequently, within a study we expect little bias arising from the selection of participants, classification of the intervention, or deviation from the intended intervention.

It is possible that the coverage of the intervention was not even throughout the study areas, and that some parts of the study areas may tend to have more *P. vivax* and some more *P. falciparum*. However, the same environmental characteristics tend to be conducive to both malaria species since they are transmitted by the same mosquito species, and the intention of the implementations were in most cases to provide the intervention to every resident.

There may be a bias in the proportion of infections that are patent by microscopy for the two species. *P. vivax* tends to have lower densities in the peripheral blood however the relative dynamics of sub-patent infections following an intervention in the two species are not well known.

There is likely to have been selection in which interventions were used in which settings. For example, MDA studies tended to be set in highly seasonal settings.

Since the primary aim of the studies was not to assess the relative impact, it is unlikely that selective reporting played a substantial role. A major reason for excluding studies was the lack of information on the intervention, particularly the timing, but it is not obvious that this would lead to bias in the relative impact.

## Series of time-points with ITN

**Clinical Cases.**    There were eleven studies [7,22,25,26,42–53] providing data on clinical malaria cases when a first-time distribution of ITNs took place, providing 26 series of time-points and 13 series of data points from six studies [7,22,43,45,46,51,54–56] in areas with repeated ITN distributions, where there had been a previous distribution or high coverage of bed nets (Fig 3).

**Table 3. Characteristics of the 69 series of time-points of patent infections by intervention.**

| | ITN (first distribution) | ITN (repeated distribution) | MDA (first round) | IRS (first spraying) | IRS (repeated spraying) | IRS (undetermined round) | Overall |
|---|---|---|---|---|---|---|---|
| **Age group** | | | | | | | |
| 0–15 years | 3 (20%) | 2 (8%) | 5 (50%) | 1 (50%) | 5 (100%) | 10 (91%) | 26 (38%) |
| 2+ years | 0 (0%) | 1 (4%) | 0 (0%) | 0 (0%) | 0 (0%) | 0 (0%) | 1 (1%) |
| 5+ years | 0 (0%) | 0 (0%) | 1 (10%) | 0 (0%) | 0 (0%) | 0 (0%) | 1 (1%) |
| 15+ years | 1 (7%) | 0 (0%) | 0 (0%) | 0 (0%) | 0 (0%) | 0 (0%) | 1 (1%) |
| all ages | 11 (73%) | 23 (88%) | 4 (40%) | 1 (50%) | 0 (0%) | 1 (9%) | 40 (58%) |
| **Coverage[1]** | | | | | | | |
| low | 7 (47%) | 4 (15%) | 5 (50%) | 0 (0%) | 0 (0%) | 0 (0%) | 16 (23%) |
| high | 7 (47%) | 20 (77%) | 4 (40%) | 1 (50%) | 5 (100%) | 11 (100%) | 48 (70%) |
| missing | 1 (7%) | 2 (8%) | 1 (10%) | 1 (50%) | 0 (0%) | 0 (0%) | 5 (7%) |
| **Seasonality[2]** | | | | | | | |
| low | 11 (73%) | 24 (92%) | 4 (40%) | 2 (100%) | 0 (0%) | 8 (73%) | 49 (71%) |
| high | 4 (27%) | 2 (8%) | 6 (60%) | 0 (0%) | 5 (100%) | 3 (27%) | 20 (29%) |
| **Relapse pattern** | | | | | | | |
| frequent | 12 (80%) | 25 (96%) | 6 (60%) | 2 (100%) | 0 (0%) | 0 (0%) | 45 (65%) |
| long | 0 (0%) | 0 (0%) | 1 (10%) | 0 (0%) | 0 (0%) | 0 (0%) | 1 (1%) |
| both | 3 (20%) | 1 (4%) | 3 (30%) | 0 (0%) | 5 (100%) | 11 (100%) | 26 (33%) |
| **Transmission[3]** | | | | | | | |
| low Pf low Pv | 3 (20%) | 12 (46%) | 0 (0%) | 0 (0%) | 1 (20%) | 5 (45%) | 21 (30%) |
| low Pf high Pv | 5 (33%) | 0 (0%) | 2 (20%) | 0 (0%) | 4 (80%) | 6 (55%) | 17 (25%) |
| high Pf low Pv | 1 (7%) | 12 (46%) | 1 (10%) | 0 (0%) | 0 (0%) | 0 (0%) | 14 (20%) |
| high Pf high Pv | 6 (40%) | 2 (8%) | 7 (70%) | 2 (100%) | 0 (0%) | 0 (0%) | 17 (25%) |
| **Diagnostic tool** | | | | | | | |
| microscopy | 10 (67%) | 12 (46%) | 8 (80%) | 2 (100%) | 5 (100%) | 11 (100%) | 48 (70%) |
| PCR | 3 (20%) | 12 (46%) | 2 (20%) | 0 (0%) | 0 (0%) | 0 (0%) | 17 (25%) |
| microscopy and PCR | 2 (13%) | 2 (8%) | 0 (0%) | 0 (0%) | 0 (0%) | 0 (0%) | 4 (6%) |
| **Type of study** | | | | | | | |
| observational | 12 (80%) | 26 (100%) | 10 (100%) | 1 (50%) | 5 (100%) | 9 (82%) | 63 (91%) |
| trial | 3 (20%) | 0 (0%) | 0 (0%) | 1 (50%) | 0 (0%) | 2 (18%) | 6 (9%) |
| **Survey taken in different seasons[4]** | | | | | | | |
| yes | 6 (40%) | 3 (12%) | 9 (90%) | 0 (0%) | 0 (0%) | 1 (9%) | 19 (28%) |
| no | 9 (60%) | 2 (8%) | 1 (10%) | 1 (50%) | 5 (100%) | 10 (91%) | 28 (41%) |
| unknown | 0 (0%) | 21 (81%) | 0 (0%) | 1 (50%) | 0 (0%) | 0 (0%) | 22 (32%) |
| **Time points per time series** | | | | | | | |
| mean | 2.8 | 2.3 | 5.6 | 3.5 | 2 | 2.1 | 2.9 |
| median (min,max) | 2 (2,6) | 2 (2,5) | 4.5 (2,13) | 3.5 (2,5) | 2 (2,2) | 2 (2,3) | 2 (2,13) |

[1]High coverage is defined as more than 70% of people with access to intervention for ITN and 90% of people for IRS. For MDA high coverage is defined as more than 90% receiving at least one treatment

[2]Seasonality is defined using the measure of relative entropy described by Pascale (2015) [29] which quantifies the distribution of rainfall across the year

[3]Transmission intensity cut off was at 5% prevalence for the respective species or less than 100 cases per 1000 people per year

[4]The data collection may spread across different seasons or all the data points described may be collected throughout the same season (e.g. always during the wet season)

For the first-time distributions of ITNs, the proportion of cases that were caused by *P. vivax* tended to increase for most of the series (at least initially), but there were four (15% e), g), h), l)) that remained stable and five (19% a),d),i),m),y)) that decreased. Overall, there is substantial

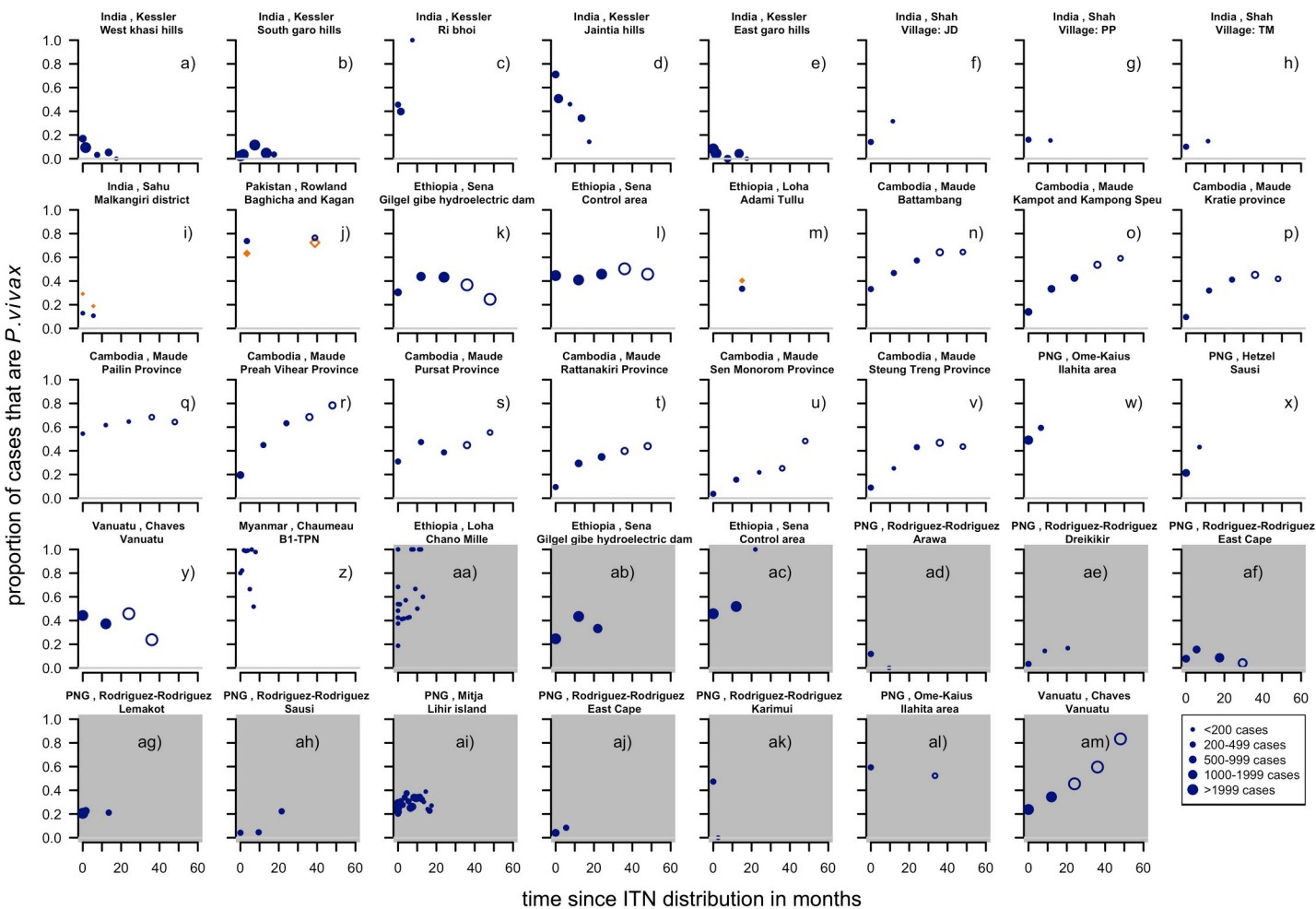

**Fig 3. Proportion of clinical cases with *P. vivax* infection by time since first ITN distribution (white background) or repeated ITN distribution (grey background).** Blue filled circles: proportion of cases with *P. vivax* infection up to 24 months from the intervention. Open circles: over 24 months after the intervention happened (excluded in the analysis because focus was on the dynamics in the first 2 years after the intervention). Orange diamonds: the proportions in the control groups. Dots at time-point 0 were not necessarily measured at this time but at time-points before the intervention (median 2.2 months before implementation, range 0–24 months before) and it was assumed that this represents the proportion just before the intervention was implemented (Methods, Definitions of variables, Time since implementation). Dot size depends on number of cases found.

variability across the different settings. The study by Maude [50] included series in several provinces in Cambodia, where the proportions tended to increase for the first 24 months (Fig 3N–3V). Increases can also be seen in Papua New Guinea (Hetzel [42], and Ome-Kaius [7]). In contrast, a steep downward trend is visible in a series (Fig 3D) from Meghalaya in the North East of India. Some of the studies had other interventions in place (S2 Supporting Information) such as IRS. In some areas there were already nets, however, no previous distribution was mentioned and the coverage was not reported to have been high. Of the 13 series, that received a repeated distribution of ITNs six (46% aa),af),ag),ai),aj),al)) showed little or no change, five (38% ab),ac),ae),ah)am)) increased, whereas two (15%,ad),ak)) showed a downward trend (Fig 3). One series from Vanuatu shows a clear upward trend (Fig 3am)). In this case, the second distribution happened three years after the first one (the first one is shown in Fig 3Y). In the first distribution, the ITN coverage was around 25% whereas the second it was 50% of individuals [45]. Some studies had spraying or vector breeding site reduction in place [55,56,51] (S2 Supporting Information). The time difference between the previous and the

repeated distributions could vary (varied from approximately three to six years after previous distribution) which means nets from previous studies could have different remaining levels of protection.

**Patent infections.** For the first-time ITN distributions, a total of nine studies [5,7,22,25,26,42,43,52,53,57–60] provided information on patent infections. This led to 15 series of which twelve were observational before and after studies. Three series provide information on control groups, of which two also report patent infections before the intervention was implemented. The studies with a first-time distribution of ITNs were located in Papua New Guinea, Cambodia, Myanmar, Solomon Islands, Afghanistan and Pakistan. Eleven (73%) show a clear increase in the proportion of patent infections that were *P. vivax* (Fig 4) three studies (20%) show a decrease and one (7%) there was little change.

For the repeated distributions, there were 26 series of time-points from seven studies [5,7,22–24,43,61–66] which were mostly located in Papua New Guinea, with single studies in Cambodia, on the Thai-Myanmar border and in Ethiopia. Of the 26 series of time-points, twelve (46%) show

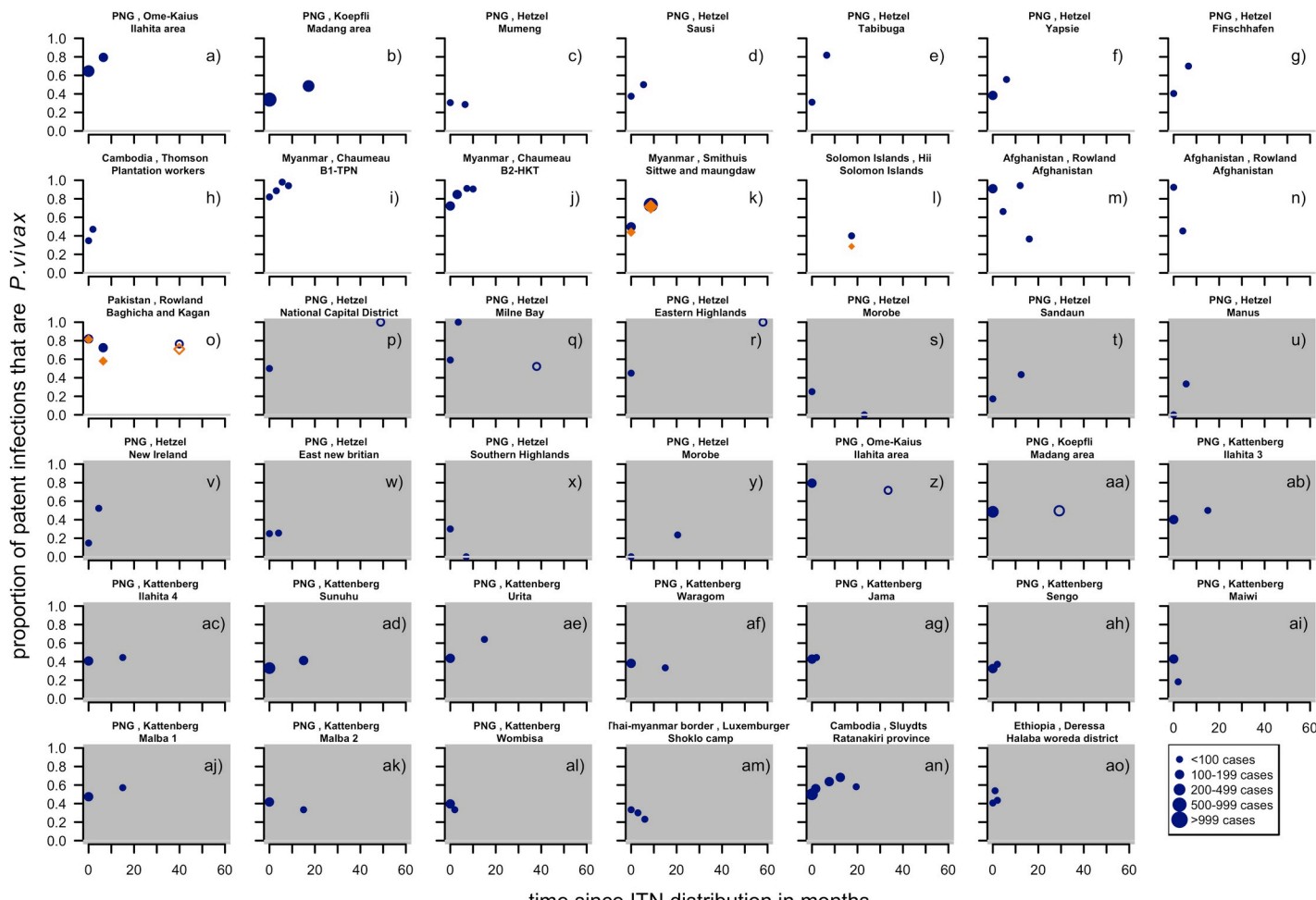

**Fig 4. Proportion of patent infections that are *P. vivax* by time since first ITN distribution (white background) or repeated ITN distribution (grey background).** Blue filled circles: proportion of cases attributed to *P. vivax* up to 24 months from the intervention. Orange diamonds: the proportions in the control groups. Dot size depending on number of infections found. Study p), r) and aa) are excluded from the regression analyses due to having no data collected in the first 24 months following the intervention. Pre-intervention points as described in Methods (definition of variables: time since implementation).

an upward trend, eight (31%) a downward trend and six (23%) show no obvious change (Fig 4). Fig 4K) and 4O) suggest that there are only slight differences in the patterns if the proportion of patent infections of the intervention group is compared to the control group instead of a before and after comparison. However, only two studies were available with both before and after data as well as a control group and this could not be investigated further.

### Series of time-points with MDA

**Clinical cases.**   There were five studies [52,53,67–70] with MDA providing 17 series of time-points, of which 13 were first-time administrations and four had received MDA three to six months previously. The studies were located in Brazil, Nicaragua, India and Myanmar. All studies were observational. Upward trends are observed in two series (12%, k) and p)) (Fig 5). For the remaining studies, the observed trends are either non-linear or unclear (41%, a), b),c), e), f), g) and l)).

Different drugs were used in the studies. Some used primaquine (S2 Supporting Information), however, the doses were not sufficient to kill hypnozoites and prevent relapses. The recommendation by the WHO to prevent relapses is to treat people for 14 days with a dose of 0.25–0.5mg/kg per day [71]. Therefore with the drug regimens used only blood stage parasites were cleared. The prophylactic period of the drugs varied. Some studies gave drugs on several occasions over three months. Most of the studies had other interventions in place in these areas (S2 Supporting Information).

**Patent infections.**   There were ten series of time-points from six studies [52,53,68,72–76] providing data on MDA and patent infections. All the series report on first time distributions of MDA and were observational. They were located in Brazil, Papua New Guinea, Myanmar, Indonesia and Palestine. There were eight studies with an initial upward and two studies with initial downward observed trends (Fig 6). The only study with sufficient primaquine to prevent

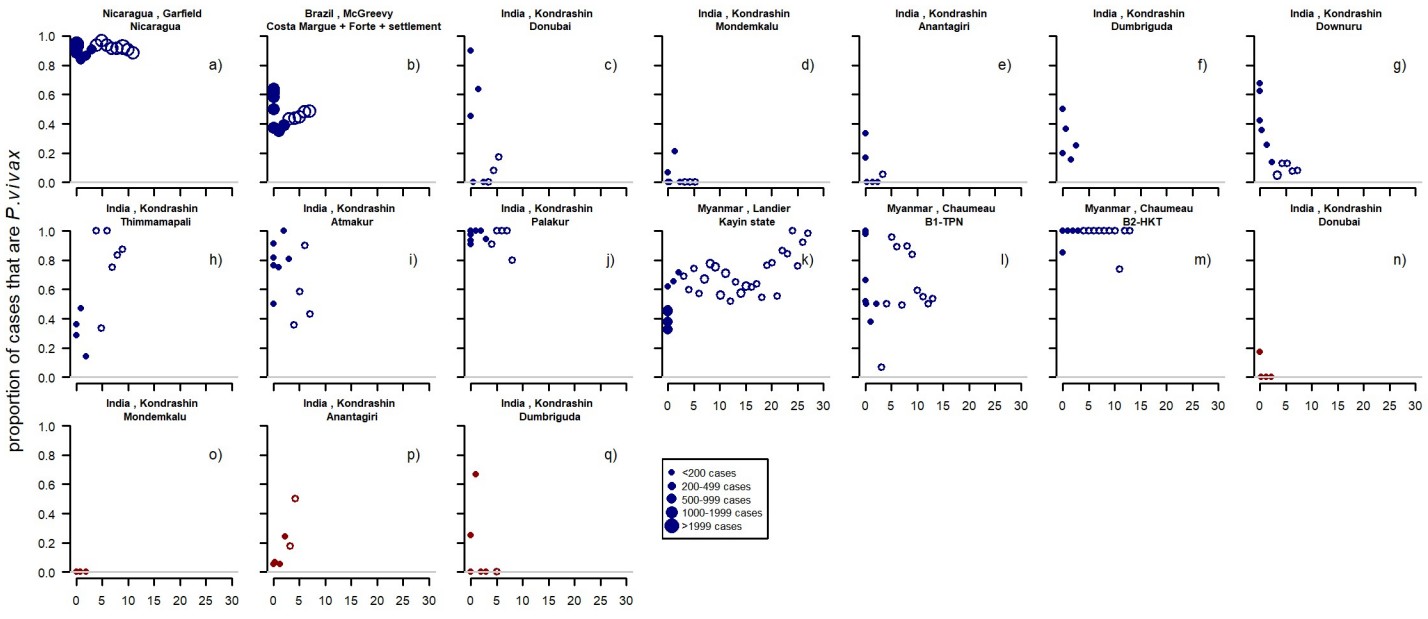

**Fig 5. Proportion of clinical cases with *P. vivax* infection by time since MDA.** Blue dots: first round of MDA, red dots: repeated MDA (first round took place 3–6 months before). Open circles: more than 3 months after MDA. Dot size depending on number of cases found Pre-intervention points as described in Methods (definition of variables: time since implementation).

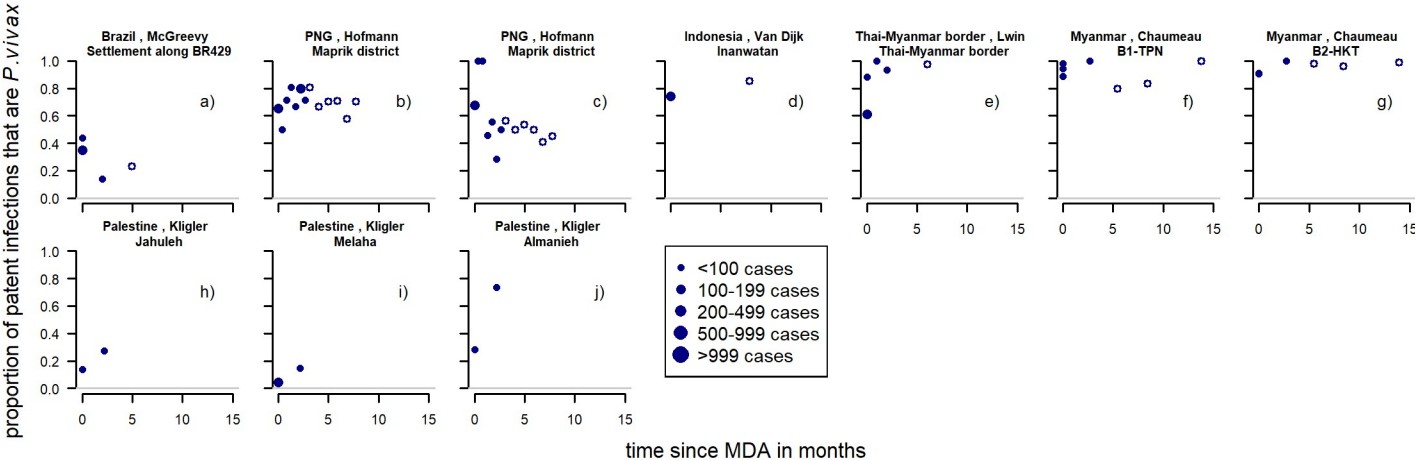

**Fig 6. Proportion of patent infections that are *P. vivax* by time since MDA.** Unfilled dots: more than 3 months after mass drug administration. Dot size depending on number of infections found Pre-intervention points as described in Methods (definition of variables: time since implementation).

relapses was by Hofmann [72] (Fig 6C)) with treatment with primaquine for twenty days with a dose of 0.5mg/kg/day in children aged 5 to 10 years. The study conducted by Kliger [74] treated with plasmochine and quinine sulphate but the dose was lower than that needed to kill the hypnozoites in these settings [77]. The study by Van Dijk [75] did 11 rounds of mass drug administration measuring patent infections before and after 10 rounds.

## Series of time-points with IRS

**Clinical cases.**   There was a total of 16 series of time-points with information on any round of IRS from eleven different studies [40,47,56,78–85]. The studies were located in Venezuela, Ethiopia, India, Vietnam, Brazil and Pakistan. All apart from two were observational studies. Five series show at least slight initial downward patterns (31%, a), d), e), j), k)) (Fig 7). Upward patterns can be observed in three studies (19%, i) o),p)). The remaining eight studies (50%) show little change or have a large amount of stochasticity. For four series, there had been no previous IRS. For nine series, there was no spraying mentioned previous to the study however, it was also not clearly indicated that there was none. In the study by Doke et al [81], there was previous spraying with DDT but due to resistance this was changed to lambdacyhalothrin. Various insecticides were used in the studies (S2 Supporting Information).

**Patent infections.**   There were 18 series of time-points from six studies [60,84–88], of which 16 were from observational studies and two had control groups. The studies were located in Indonesia, Pakistan, the Solomon Islands and India. For two series it was the first spraying, for five it was likely that spraying had happened before, and for the remaining studies it was unclear. There were nine series of time-points that showed a clear upward trend in the proportion of cases attributed to *P. vivax* (Fig 8). Five series of time-points show downward trends and the remaining four studies show little (initial) change.

## Graphical summaries of the proportion of cases or infections that were *P. vivax* over time by factor

Overall, there was substantial variability in the patterns over time. We used spaghetti plots to investigate visually if there are general patterns by each factor. Other characteristics of the series are described in S1.

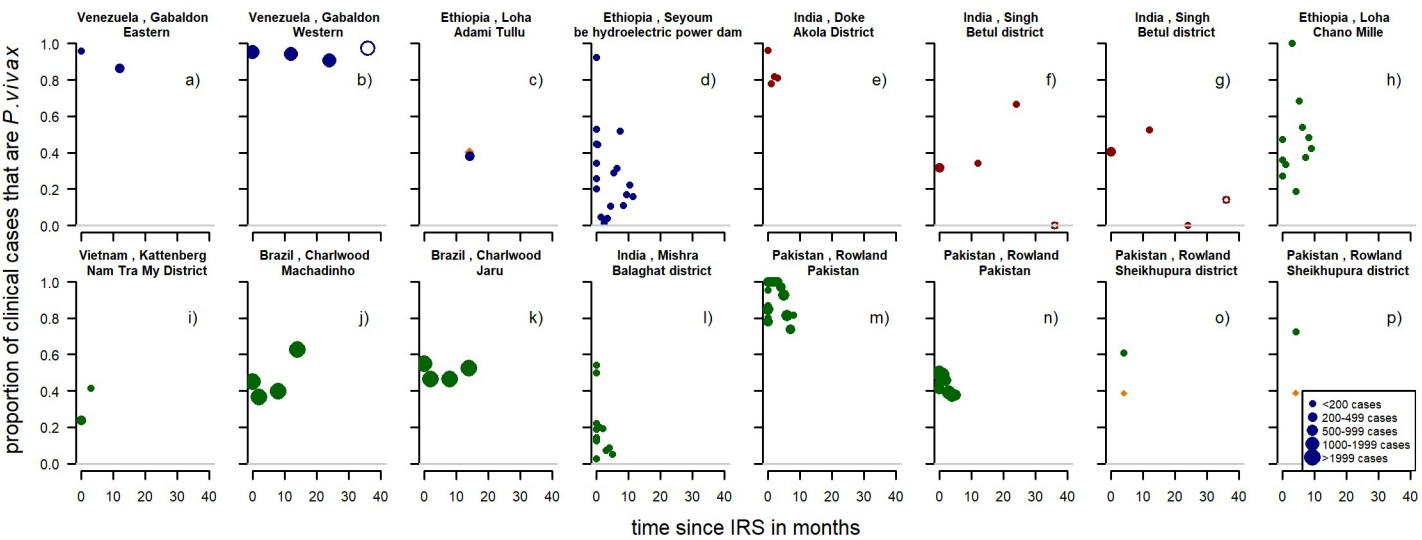

**Fig 7. Proportion of clinical cases with *P. vivax* infection by time since IRS.** Blue: first time spraying (there was no spraying for at least a year before), red: repeated spraying (spraying had occurred previously but there was a change in insecticide used), green: unclear if spraying happened before (no mention of spraying before the study but also no clear mentioning of spraying being the first time). Open circles: more than 24 months after IRS, Orange diamond shaped dots are from the control group. Dot size depending on number of cases. Pre-intervention points as described in Methods (definition of variables: time since implementation).

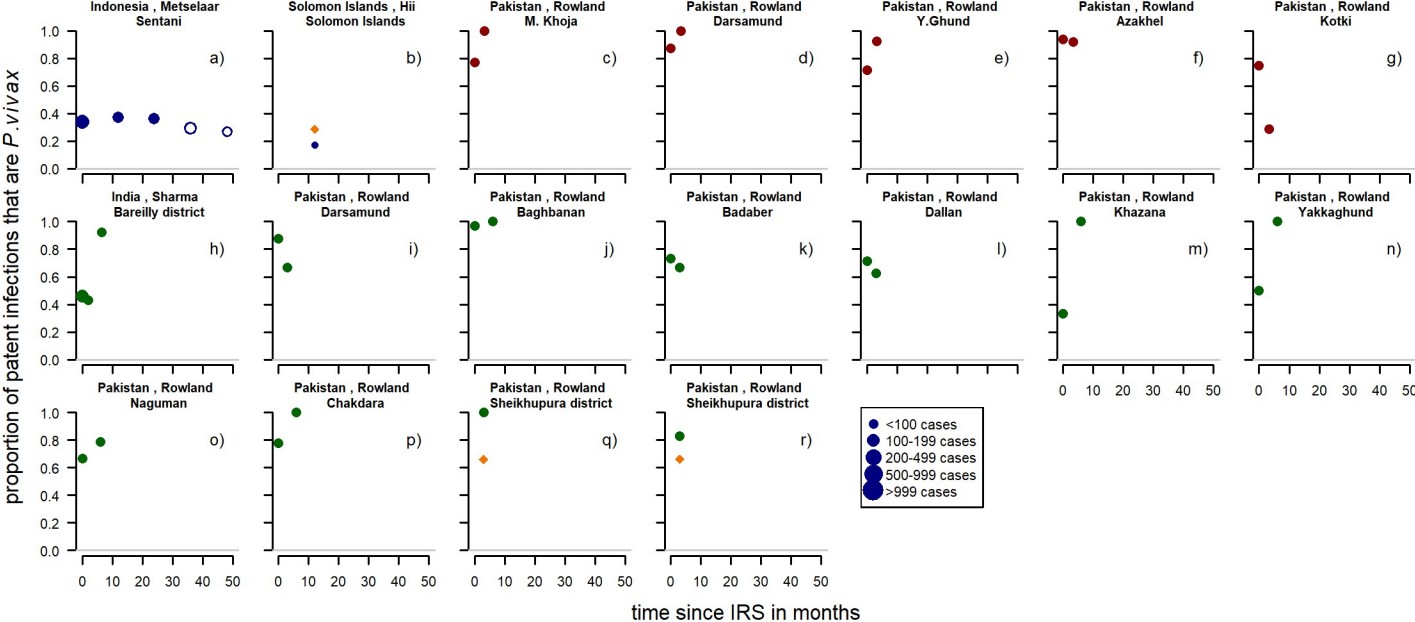

**Fig 8. Proportion of patent infections that are *P. vivax* by time since IRS.** Blue: first time spraying (there was no spraying for at least a year before), red: repeated spraying (spraying happened before but there was a change in insecticide used in panel f) and g)), green: unclear if spraying happened before (no mentioning of spraying before the study but also no clear mentioning of spraying being the first time). Open circles: more than 24 months after IRS, Orange diamond shaped dots are from the control group. Dot size depending on number of cases found. Pre-intervention points as described in Methods (definition of variables: time since implementation).

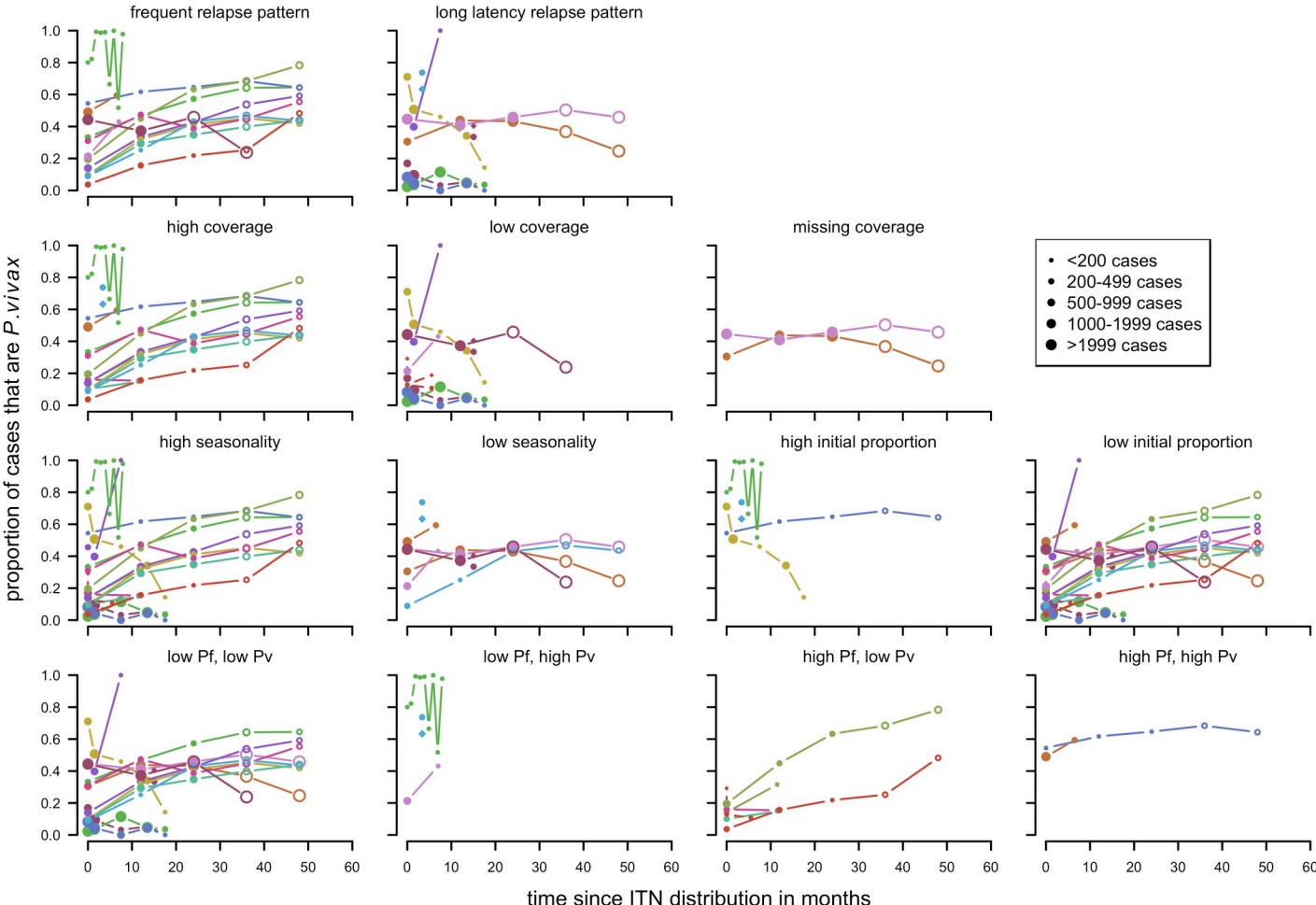

**Fig 9. Spaghetti plots splitting the series of time-points of the proportion of cases with *P. vivax* infection by relapse pattern, coverage, seasonality, initial proportion of *P. vivax* and transmission intensity after first time ITN distribution.** Dot size depending on number of infections.

**Series with ITN by factor.**    With first time distribution of ITNs with clinical cases, there were different trends observed over time by relapse patterns with a stronger tendency for upward trends in the frequent relapse category (Fig 9). The same is true for high coverage, however, many of the studies located in frequent relapse pattern areas also received interventions with high coverage. No clear differences were observed for the other factors. Age and transmission intensity could not be investigated since all but one studies looked at all ages apart from Ome-Kaius [7] (Fig 3W) with under 5-year-olds. Most studies were classified as having low transmission intensity for both *P. falciparum* and *P. vivax* with prevalence rates lower than 5% (or if prevalence was not available incidence lower than 100 cases per 1000 people per year was considered low).

For the repeated distribution of ITN with clinical cases, no clear patterns were observed for any of the factors (Fig 10). Some factors could not be assessed due to the data available.

To consider the hypothesis that the short-term dynamics may differ following first time and repeated ITN distributions, they were presented directly in the same graph. There is substantial variation in both groups (Fig 11), with a potential tendency for a slightly greater overall increase in the proportion that were *P. vivax* for the first time ITN distributions.

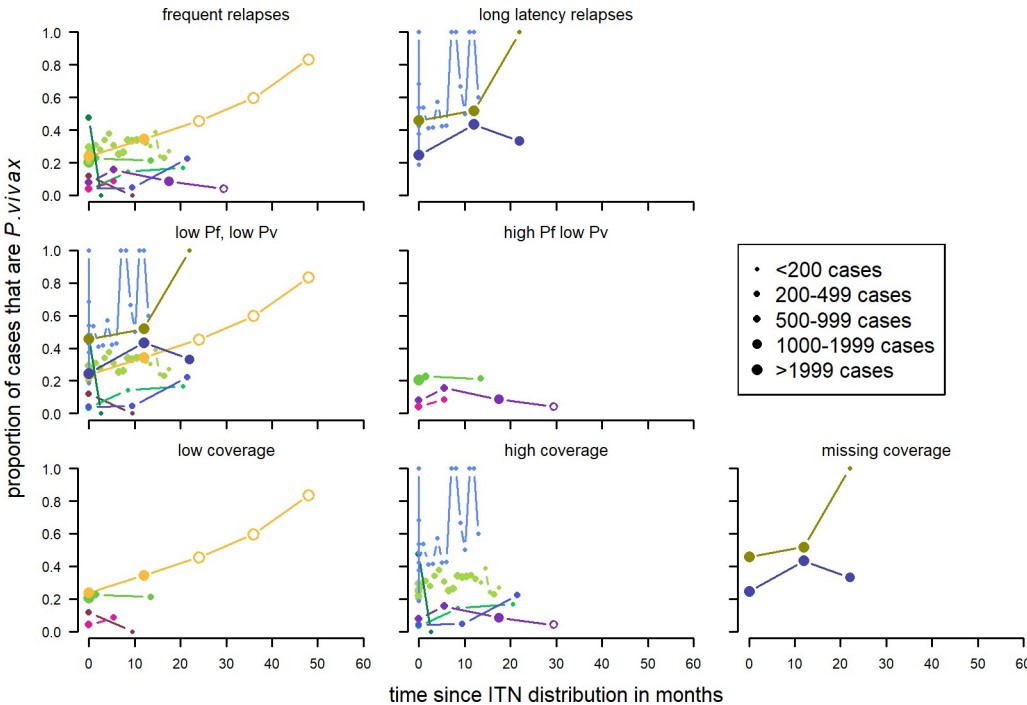

**Fig 10. Spaghetti plots splitting the series of time-points of the proportion of cases with *P. vivax* infection by relapse pattern, transmission intensity and coverage after repeated ITN distribution.** Excluding series of time points that have no measurement within 24 months after the intervention. Dot size depending on number of infections.

For a first-time distribution of ITN with patent infections, visually the variables with the most distinct patterns between categories were relapse patterns and coverage (Fig 12). When there was a repeated distribution with patent infections, none of the factors had obvious differences between categories of any factors. Relapse pattern, seasonality and transmission intensity could not be investigated due to data availability (Fig 13).

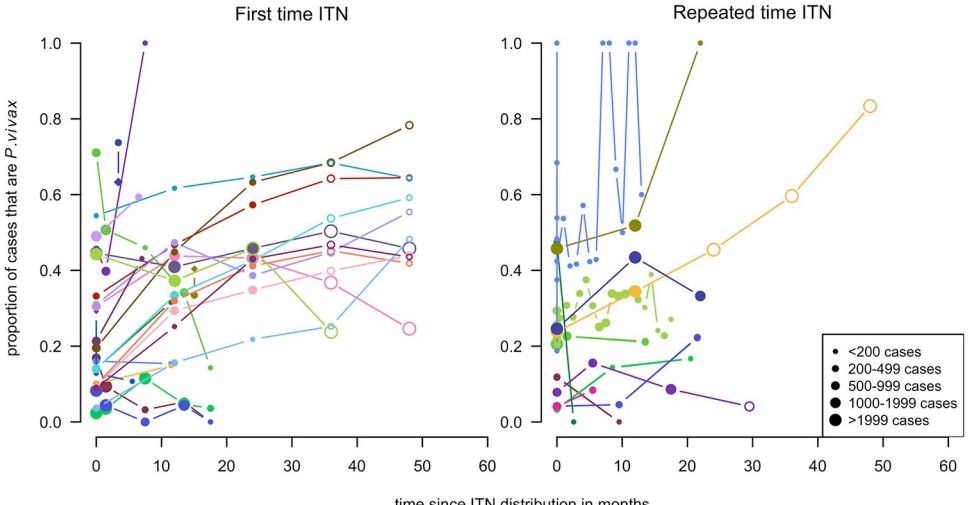

**Fig 11. Spaghetti plots splitting the series of time-points of the proportion of cases with *P. vivax* infection by round of ITN distribution.** Excluding series of time points that have no measurement within 24 months after the intervention. Dot size depending on number of infections.

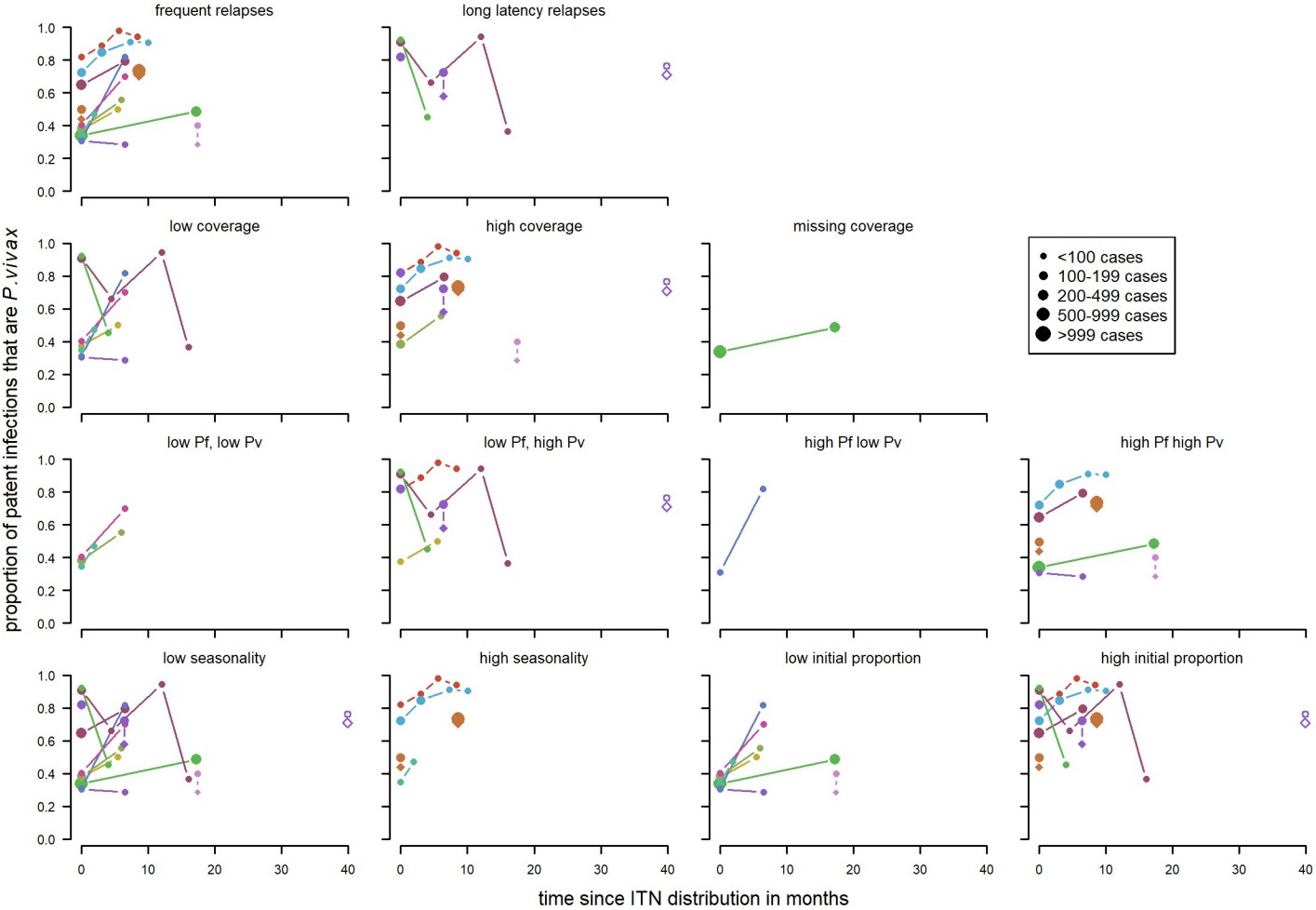

**Fig 12. Spaghetti plots splitting the series of time-points of the proportion of patent infections that are *P. vivax* by relapse pattern, coverage, transmission intensity, seasonality and initial proportion of *P. vivax* after first time distribution of ITNs.** Dot size depending on number of infections.

When plotted by distribution, there was no clear difference in the patterns over time by round of ITN distribution (Fig 14).

**Series with mass drug administration by factor.**   The first and repeated rounds of MDA were not separated for the analysis because MDA is assumed to act most strongly in the short-term and therefore similar effects were expected for first and repeated rounds of MDA.

**Clinical cases:** Most of the spaghetti plots (Fig 15) show no clear patterns in the proportion of clinical cases that were *P. vivax* by time and there was substantial variation in all categories. The studies with three rounds of mass drug administration tended to have longer follow-up times. Coverage was considered high for all studies where 90% or more of people received at least one round of MDA. There appears to be an upward trend in the proportion of cases attributed to *P. vivax* with high coverage, for studies receiving three rounds of MDA, frequent relapse pattern areas as well as low *P. falciparum* and high *P. vivax* transmission. However, it is mostly the same studies for the categories for all of these variables. The time-period of the study may also be a potential factor but there were only three studies later than the year 2000

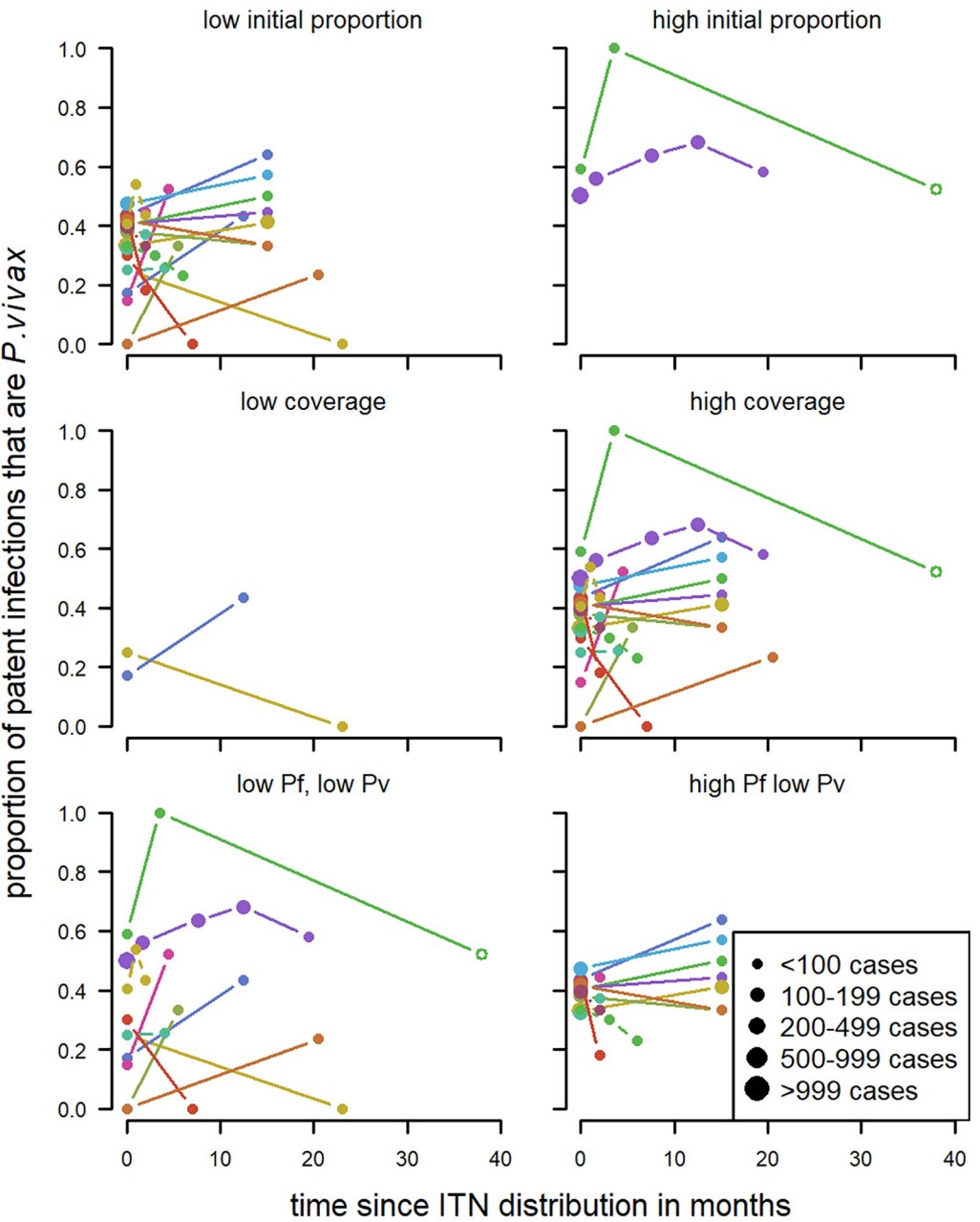

**Fig 13. Spaghetti plots splitting the series of time-points of the proportion of patent infections that are *P. vivax* by initial proportion of *P. vivax*, coverage and transmission intensity after repeated distribution of ITNs.** Excluding series of time points that have no measurement within 24 months after the intervention. Dot size depending on number of infections.

and these tended to have higher coverage, three rounds instead of one, and longer follow-up periods. Migration from areas with a high malaria transmission intensity was not captured in the data but was given as one possible explanation for the studies in India by Kondrashin et al. [69] for the areas where *P. vivax* decreases but *P. falciparum* does not.

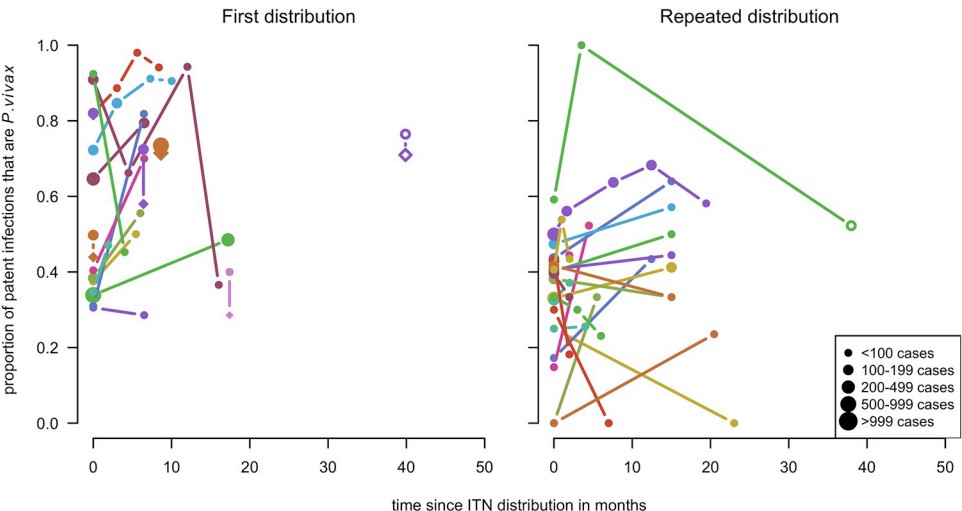

**Fig 14. Spaghetti plots splitting the series of time-points of the proportion of patent infections that are *P. vivax* by round of ITN distribution.** Excluding series of time points that have no measurement within 24 months after the intervention. Dot size depending on number of infections.

**Patent infections:** Overall there were no clear patterns in the proportion of cases with *P. vivax* parasitemia (Fig 16) apart from series in long latency relapse pattern areas where there appears to be an increase. Other factors such as rounds of MDA were not considered as all of the studies had three rounds of MDA or more in total every month apart from Hofmann *et al* [72] with one round over twenty days and McGreevy *et al* [68] where the number of rounds was not recorded.

**Series with indoor residual spraying by factor.**    For indoor residual spraying, for both clinical cases (Fig 17) and patent infections (Fig 18), no clear visual patterns for any of the variables emerged. There was a substantial amount of variation.

**Regression analyses to estimate the observed trends in the proportion of *P. vivax* cases by time since intervention and identify potential factors.**    To estimate the changes over time since the intervention and to identify factors that are associated with the change over time, logistic regression was used with random effects for the intercept and slope for series (S3 Supporting Information). When possible, models included several explanatory variables, however, since the number of series for each level of the covariates is limited and an interaction term for the effect of the covariates is needed to estimate the effect over time the full models did not always converge. In those cases, the analysis was carried out for one factor at a time.

The analyses were done separately for the interventions as the effects of the explanatory variables might differ by intervention. Adjustment for the season at the time of the surveys was included in all models.

In summary, for all interventions there is substantial variability in the trends of the proportion of clinical cases or patent infections that were *P. vivax* in the short-term by time since the implementation (Tables 4 & 5). The time-points used were up to 24 months after the intervention for ITN and IRS and up to three months for MDA. Within these time periods, the patterns were roughly linear.

When the interventions are compared overall, there is a tendency for upward patterns in malaria attributed to *P. vivax* for ITN and IRS, with MDA estimates uncertain in the first three months, although these associations are not significant. There was no evidence that the round

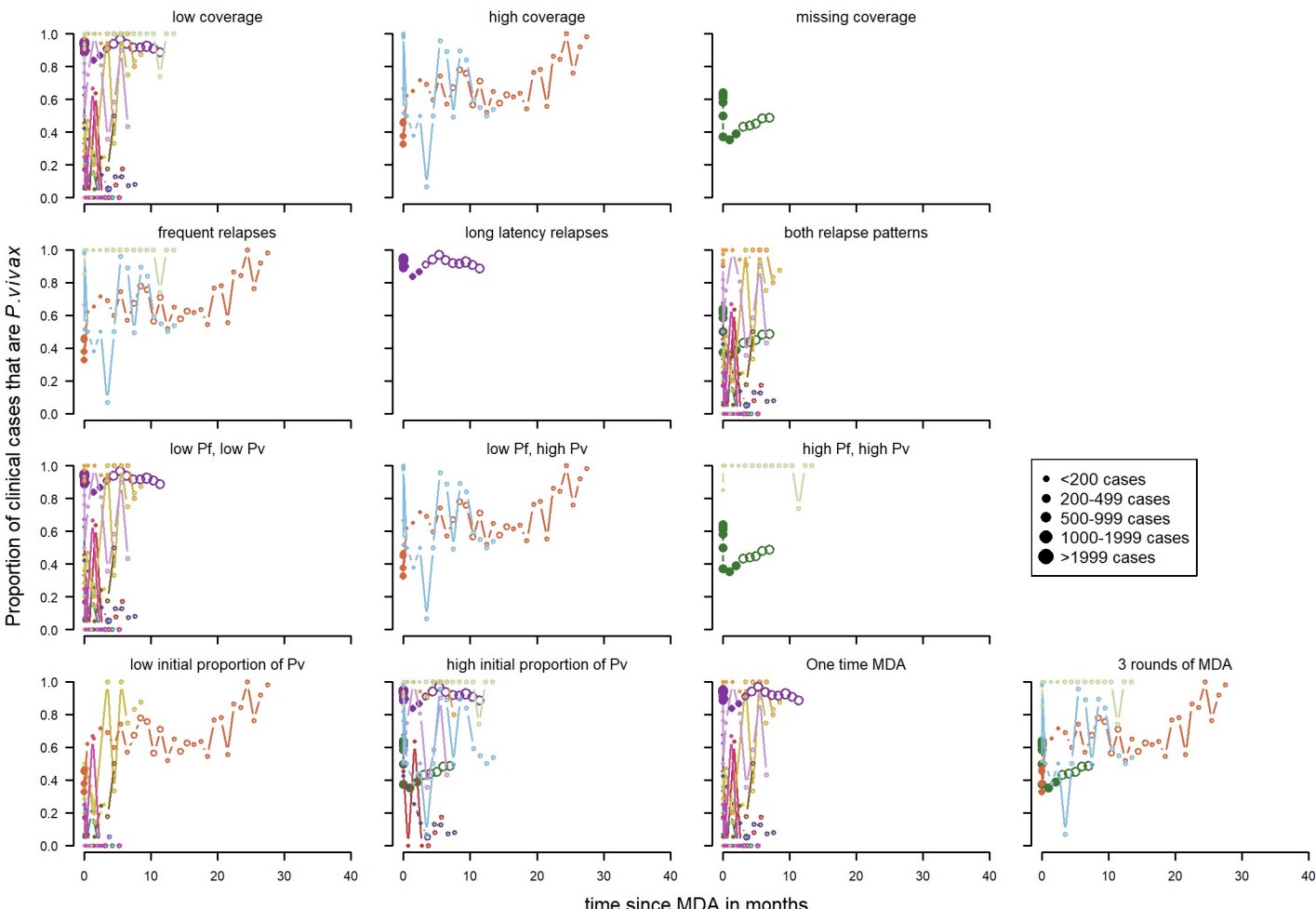

**Fig 15. Spaghetti plots splitting the series of time-points of the proportion of cases with *P. vivax* infection by rounds of MDA, coverage, relapse pattern and transmission intensity after MDA.** Dot size depending on number of infections.

of intervention for ITN differs significantly in a model with time however the effects of the explanatory variables differ by round of distribution. Overall, the analyses were unable to rule in or rule out explanatory factors due to the limited number of series and the collinearity of some variables. There was evidence of an effect of coverage for some of the interventions, however not all of them and this could not be investigated for IRS. Transmission intensity had significantly different effects by species for ITN (Fig 19) but no clear pattern was observed for MDA and IRS. The results suggest that relapse pattern may be able to explain some of the variation with long latency patterns tending to lead to a weaker increase in *P. vivax* for ITN as well as for IRS. There is uncertainty in case of seasonality as it could not be tested for all interventions but possibly had an effect in others. The initial proportion of *P. vivax* is a possible factor, however, this is uncertain and it was not signficant in most analyses.

## Discussion

To our knowledge, this is the first systematic review of the relative effect of interventions on sympatric *P. vivax* and *P. falciparum*. The mean increase in the proportion of cases that were

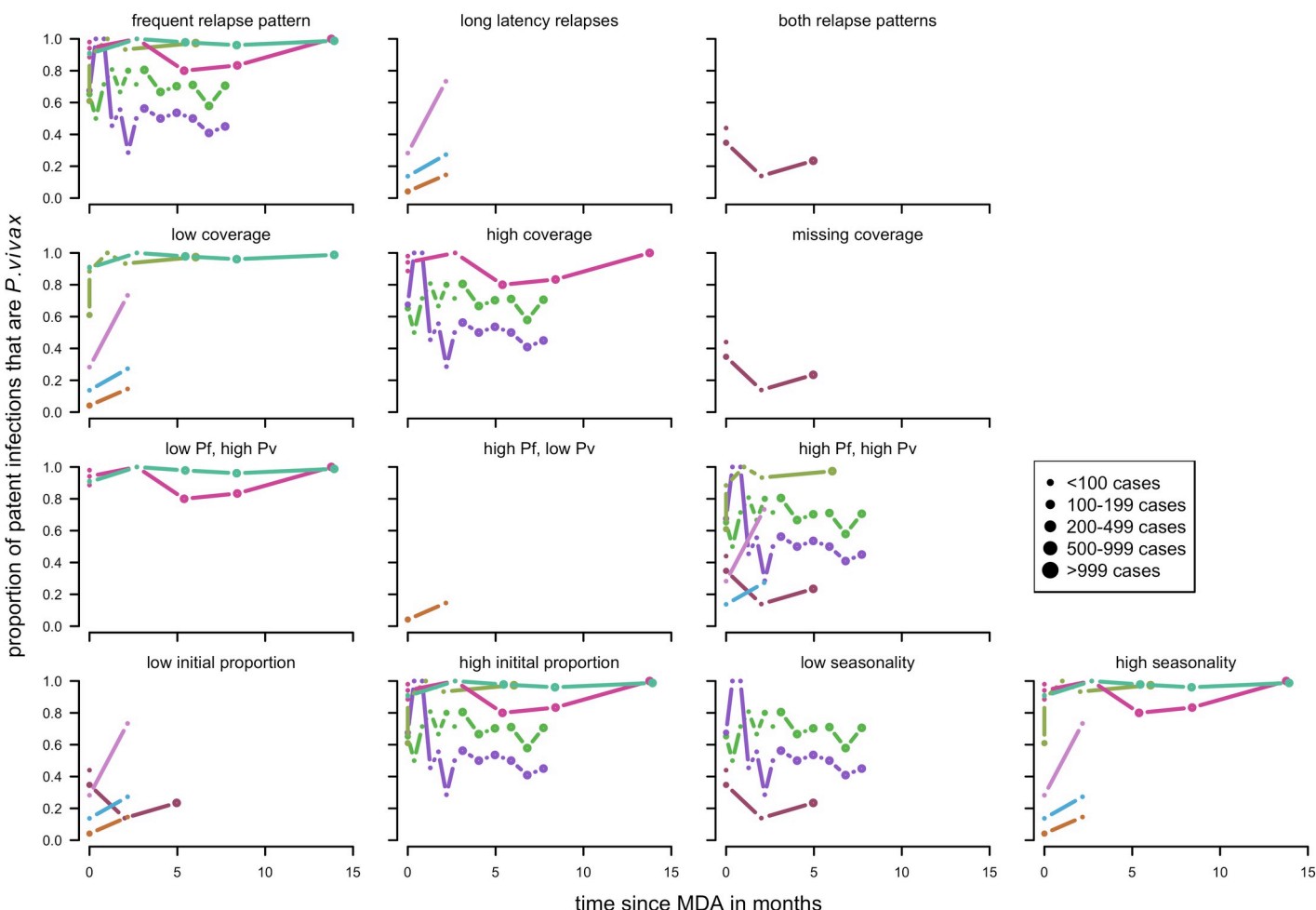

**Fig 16. Spaghetti plots splitting the series of time-points of the proportion of patent infections that are *P. vivax* by relapse pattern, coverage, transmission intensity, initial proportion of P. vivax and seasonality after MDA.** Dot size depending on number of infections.

*P. vivax* either significantly increased or the estimates were in the direction of an increase following ITN, IRS and for MDA after the first three months. However, the most striking finding was the substantial variability found in the relative impact for all of the interventions over the time-scale of this study: up to 24 months for ITN and IRS and up to six months for MDA.

A recent review by Price [6], found that the proportion of *P. vivax* tends to increase when malaria control interventions are implemented, over much longer time scales.

Due to the number of series, the variability in the observed trends and collinearity between variables, it was not possible to rule in or rule out many of the variables as having an impact on the trend over time. Variables which are likely to have an effect were transmission intensity, coverage and relapse pattern. Transmission intensity was either significantly associated or had large odds ratios in several, but not all, of the analyses. It is likely that transmission intensity plays a role as the effect of interventions can be different for single species for different transmission intensities and different transmission levels can pose different challenges such as more heterogeneity and low level parasitemia in low transmission areas [89].

Even though there was no evidence for an impact of coverage for all interventions, it may have an influence if an intervention does have different relative effects. High coverage may

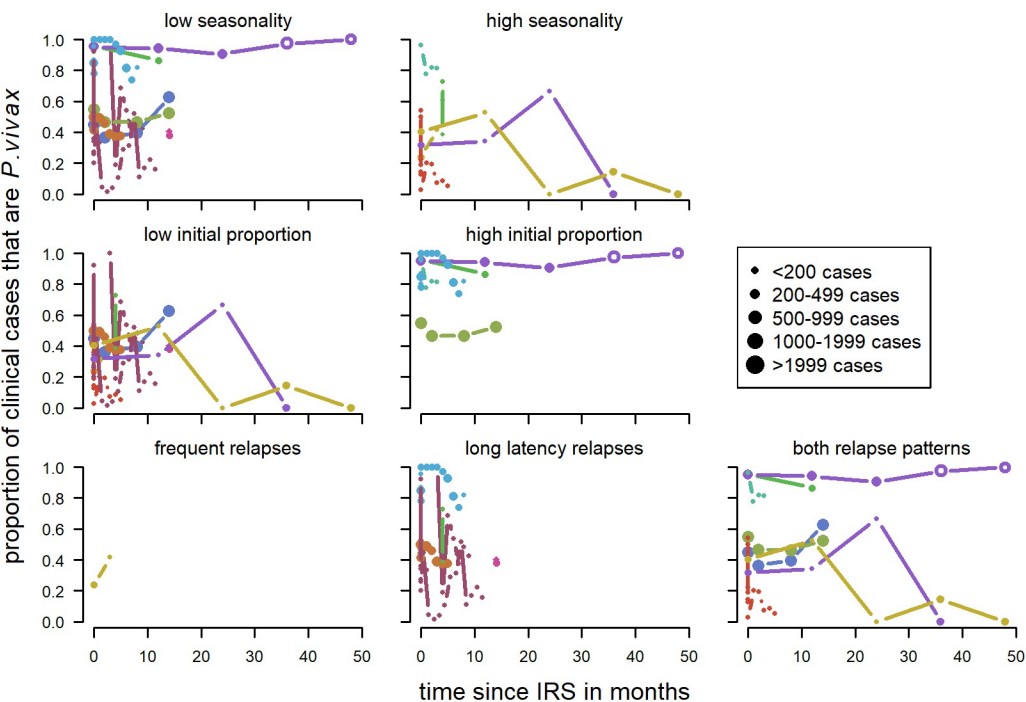

**Fig 17. Spaghetti plots splitting the series of time-points of the proportion of cases with *P. vivax* infection by seasonality, initial proportion of *P. vivax* and relapse pattern after IRS.** Dot size depending on number of infections.

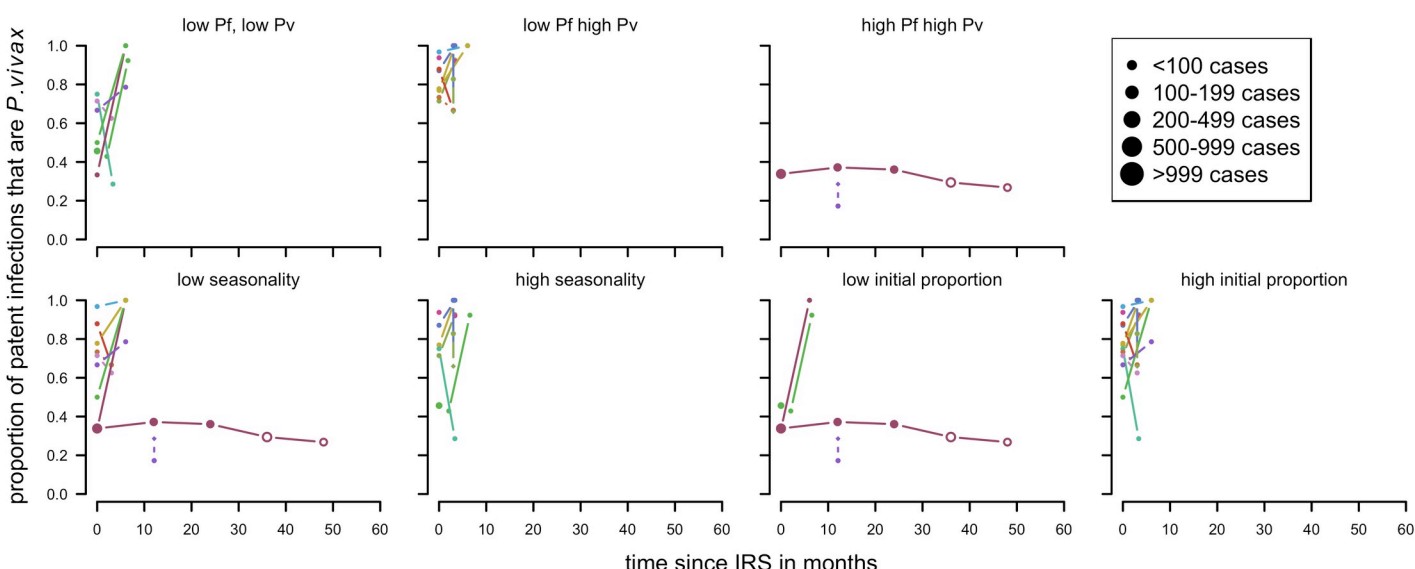

**Fig 18. Spaghetti plots splitting the series of time-points of the proportion of patent infections that are *P. vivax* by seasonality, initial proportion of P. vivax and transmission intensity after IRS.** Dot size depending on number of infections.

**Table 4. Estimated changes in the proportion of clinical cases that were *P. vivax* over time for each intervention separately, adjusted for season at time-point of survey.**

| Intervention | Time-span | Number of series of time points | Change over time: Odds ratio per month | 95% CI | Variation between series (min/max estimated odds ratio per month) |
|---|---|---|---|---|---|
| ITN (first round) | 24 months | 26 | 1.03 | 1.01–1.06 | 0.92–1.13 |
| ITN (repeated round) | 24 months | 12 | 1.04 | 1.01–1.07 | 0.99–1.10 |
| MDA | <3 months | 17 | 0.93 | 0.63–1.39 | 0.37–2.39 |
| MDA | 3–6 months | 13 | 1.62 | 1.14–2.30 | 1.08–2.06 |
| IRS | 24 months | 16 | 1.00 | 0.95–1.06 | 0.87–1.24 |

also potentially lead to a stronger reduction in transmission levels which in turn could affect levels of acquired immunity. Relapse pattern may have an effect with differences between long latency and frequent relapse pattern. The long latency relapse pattern, which has fewer relapses [27], might lead to weaker increases in the proportion of cases attributed to *P. vivax* (as was found with significance for first time ITN distribution patent infections and IRS clinical cases, and in the direction of a weaker increase though not significant for the first three months after MDA clinical cases and first time ITN distribution clinical cases). The effects of seasonality and the initial proportion of *P. vivax* were uncertain.

Other factors could be important but could not be investigated due to the limited data. There were some variables for which data was collected but there was little variety in values such as age and diagnostic test. For some of the considered factors more detailed information would have been beneficial such as previous coverage of the intervention or inequality in coverage across a community. Although the season at which data points were collected was considered, the exact timing during the season may be informative [90]. There were also some variables for which data was not collected but could have an impact, for example, drug resistance which could vary for the two species. Other factors could be migration, access to treatment, recent changes in transmission or previous interventions. The extent of competition and cross-immunity between the species is uncertain.

The results can provide information for planning and evaluating the impact of interventions. It is important to consider the biological differences between *P. vivax* and *P. falciparum* when evaluating an intervention which does not target hypnozoites. The species should be monitored separately. This would allow the adjustment of the strategy if the original intervention is less successful for one species. To aid interpretation of the observed patterns, contextual factors that could influence the malaria situation and changes in the species composition should be monitored. These factors include decreasing coverage of other interventions, human behavior such as migration, or emergence of drug resistance. There can also be changes in detection (more access to health facilities, changes in diagnostic tools) or changes in reporting (changes in case definitions, biases in selection of survey populations) [91].

More clarity could be obtained by including more studies in the analysis if they become available. However, as transmission decreases further, it may be more appropriate to employ

**Table 5. Estimated changes in the proportion of patent infections that were *P. vivax* over time for each intervention separately, adjusted for season at time-point of survey.**

| Intervention | Time-span | Number of series of time points | Change over time: Odds ratio per month | 95% CI | Variation between series (min/max) |
|---|---|---|---|---|---|
| ITN | 24 months | 37 | 1.02 | 0.98–1.06 | 0.82–1.15 |
| MDA | <3 months | 9 | 0.97 | 0.57–1.65 | 0.37–1.83 |
| IRS | 24 months | 18 | 1.27 | 1.08–1.51 | 0.93–1.75 |

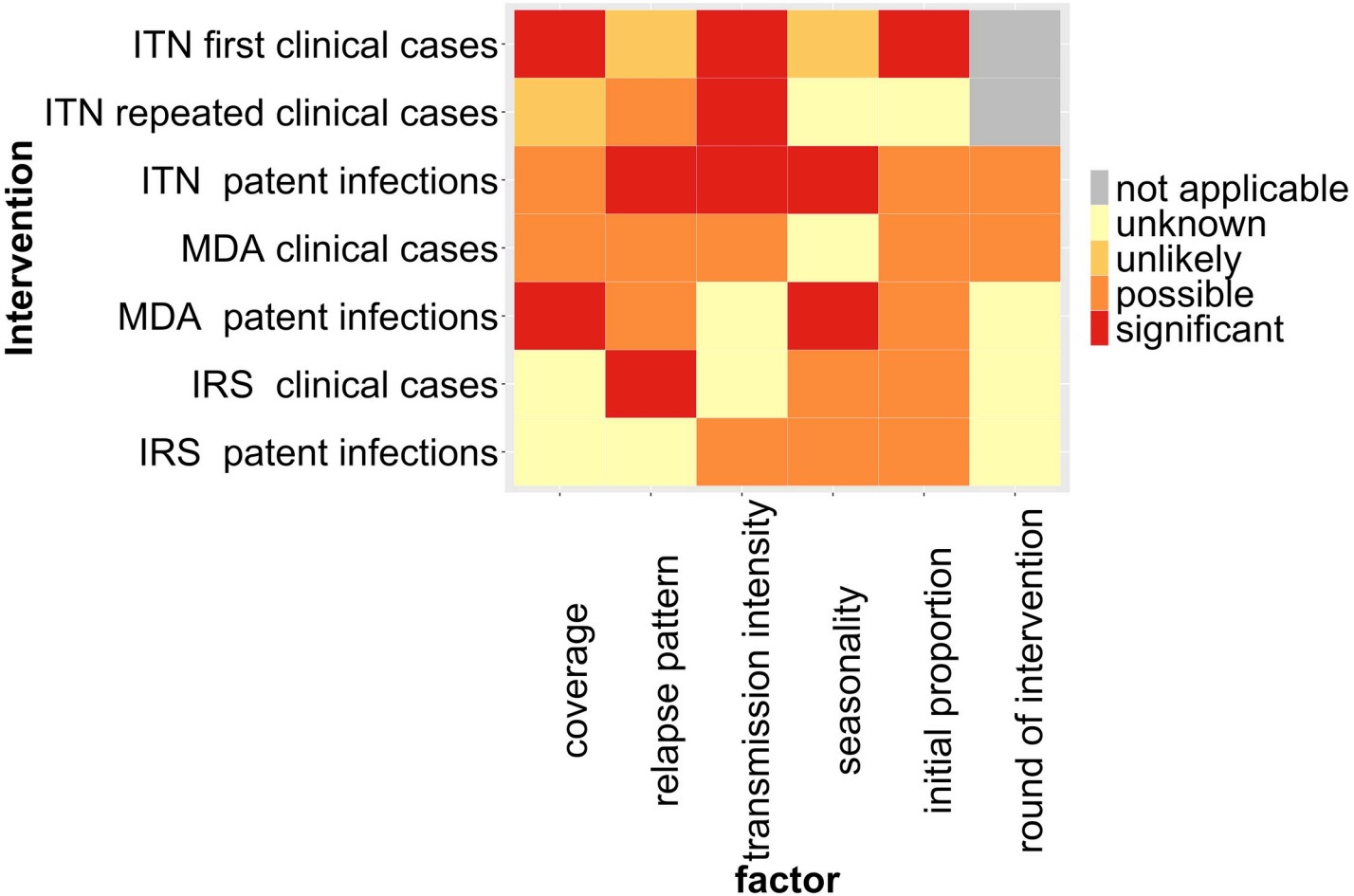

**Fig 19. Heat map of factors by intervention.** Factors were considered as 'possible' if the estimated change in odds over time was more than 1.01 per month or less than 0.99 per month in the logistic regression.

targeted or reactive interventions. It would be useful if studies describe interventions in detail, information was often lacking on the timing of the intervention as well as coverage. There were also studies that reported that there was a national control program but did not specify what this entailed and therefore could not be included in the analysis.

Where data is limited, a possible further step to gain clarity is to use modelling. Predictions can be made of the effect of variables for which there was limited data and validated models can be used to disentangle the effects of collinear covariates.

## Conclusion

When interventions are implemented in areas with sympatric *P. vivax* and *P. falciparum* malaria, there is substantial variation in how the proportions of the two species change on a short-term time scale, with a tendency for the proportion of *P. vivax* to increase overall. It was not possible to rule in or out many of the potential factors, due to the high variability and the availability of data. Relapse pattern, coverage, transmission and intervention type are possible explanations contributing to this variation but they were not consistently significant. More studies with well-described intervention implementations would reduce the uncertainty.

## Supporting information

**S1 Supporting Information. Included studies and variables by series of time-points.**
(DOCX)

**S2 Supporting Information. Interventions in place at baseline.**
(DOCX)

**S3 Supporting Information. Results from the regression analyses.**
(DOCX)

## Acknowledgments

We thank Jacqueline Huber for helping with the development of the literature search strategy, and everyone within Swiss TPH who provided insights and feedback especially Manuel Hetzel, Guojing Yang and Christian Lengeler.

## Author Contributions

**Conceptualization:** Melanie Loeffel, Amanda Ross.

**Data curation:** Melanie Loeffel.

**Formal analysis:** Melanie Loeffel, Amanda Ross.

**Methodology:** Melanie Loeffel, Amanda Ross.

**Writing – original draft:** Melanie Loeffel.

**Writing – review & editing:** Melanie Loeffel, Amanda Ross.

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
