## [Decision Letter · Decision Letter 0]

9 Mar 2022

Dear Dr. Ross,

Thank you very much for submitting your manuscript "The relative impact of interventions on sympatric Plasmodium vivax and Plasmodium falciparum malaria: a systematic review" for consideration at PLOS Neglected Tropical Diseases. As with all papers reviewed by the journal, your manuscript was reviewed by members of the editorial board and by several independent reviewers. In light of the reviews (below this email), we would like to invite the resubmission of a significantly-revised version that takes into account the reviewers' comments. 

We cannot make any decision about publication until we have seen the revised manuscript and your response to the reviewers' comments. Your revised manuscript is also likely to be sent to reviewers for further evaluation.

Sincerely,

Rhoel Ramos Dinglasan

Associate Editor

Marcelo Ferreira

Deputy Editor

Reviewer's Responses to Questions

**Key Review Criteria Required for Acceptance?**

**Methods**

-Are the objectives of the study clearly articulated with a clear testable hypothesis stated?

-Is the study design appropriate to address the stated objectives?

-Is the population clearly described and appropriate for the hypothesis being tested?

-Is the sample size sufficient to ensure adequate power to address the hypothesis being tested?

-Were correct statistical analysis used to support conclusions?

-Are there concerns about ethical or regulatory requirements being met?

Reviewer #1: Objectives and methods clearly described in detail, but statistical analysis may benefit from an expert biostatistician with experience in the field.

Reviewer #2: The objectives of this study were to conduct an analysis of the published literature reporting on the effect of three malaria control interventions implemented in regions co-endemic for Plasmodium falciparum and P. vivax in order to estimate the proportions of clinical cases and patent infections attributable to P. vivax after the implementation. 

The methods used were a literature search of publications describing implementation of the three control methods; the proportion of P. vivax infections, either clinical or patent, was calculated by dividing the number of such P. vivax cases by cases of either P. vivax or P. falciparum. The team also looked at the impact on proportion of P. vivax according to different variables. 

The team is to be commended on such an undertaking, and in the reading on the manuscript, it become clear the challenges encountered with variety of data and methodologies used and in which the primary objective was not to determine proportion of P. vivax infections. 

The objectives were articulated and study design appropriate. The populations in the manuscript were clearly defined in the Supplemental sections, and appropriate for the hypotheses. The section describing the variables was very clear and well written. There are no ethical or regulatory concerns, and my background does not allow for a critical review of the statistical methods used.

One question: I saw some studies were from 40 years ago (especially in clinical cases after MDA) and/or prior to widespread implementation to ACTs (CQ used in a few of the patent studies after MDA). I would add the range of year of the studies in the main text, perhaps for each intervention as well. Did the authors consider this factor at all in their own analysis or did they notice any trends that are not reported here?

**Results**

-Does the analysis presented match the analysis plan?

-Are the results clearly and completely presented?

-Are the figures (Tables, Images) of sufficient quality for clarity?

Reviewer #1: The presentation of results is a bit tedious and overwhelming, particularly when describing figures 3 - 20. To avoid distractions and sole description of the graphs, the authors may benefit of rearranging the presentation and transferring these figures to supplemental material. It will require a little creativity to still briefly describe those results in the main text, while highlighting the overall “big picture”, which is shown in figure 21. All the other tables and boxes from the main text are important are clearly presented and should be kept as they are.

Reviewer #2: The results were complete, of good quality clearly presented, and quite informative. The Figures accurately reflected the results and the complexity thereof. In Fig 3-9 graphs, I liked the use of differently sized circles in addition to proportion of cases on Y-axis. I was curious about effects on sheer numbers.

A few minor points/edits

Figure 3. In the legend it says “Dots at time-point 0 were not necessarily measured at this time but at any time-point before the intervention…Only one pre-implementation point was available”, but in some of the graphs there are multiple points at Time zero (graph a for example). Is it the scale of the X-axis? It also states Blue filled circles: proportion of cases with P. vivax infection”. Are these numbers raw numbers of P. vivax cases or P. vivax cases divided by Pv/Pf? 

Figure 4 and 7-9. There are multiple data points at Time zero. Are there different populations within the study i.e., different villages? Or what do they generally represent?

**Conclusions**

-Are the conclusions supported by the data presented?

-Are the limitations of analysis clearly described?

-Do the authors discuss how these data can be helpful to advance our understanding of the topic under study?

-Is public health relevance addressed?

Reviewer #1: The conclusions should more clearly state that such high variability in outcomes did not identify a significant factor influencing prevalence of vivax post-intervention. "Trends" can be overseen as indicator of prediction, so words should be carefully chosen in the abstract and conclusions. Limitations were clearly described and the study will be useful for epidemiologists in the field and governing agencies responsible control measures.

Reviewer #2: The conclusions reached were that in the short term, 2 years for ITN and IRS, and MDA for >/+ 3 months, an increase in proportion of cases due to P. vivax is seen, albeit with extensive variability among studies. 

The data and analysis are helpful in highlighting the complexities of these type of studies on how the variables can effect each intervention differently, in particular the relapsing patterns, which may have more implications for a country such as India or Brazil, which may not have the homogenous fast relapsing patterns of Southeast Asia. 

The authors make a nice comparison that this is an analysis different from the long-term analysis of Price et al. The long-term effects of control on proportionality are quite evident in GMS where >75% of cases are now P. vivax in each country (except Vietnam, although there were less than 1,500 case total in 2021). The authors do state “…change in species composition should be monitored”. This point is crucial, and according to the heat map, as are patterns and transmission intensity. The authors go on to advocate for “including more studies in the analysis as they become available… and to describe interventions in detail..” This paper looks back at what has been done, but from here on out, household ITNs and IRS may no longer be the best modalities for all co-endemic countries, and MDA strategies will need to change.

 Can the authors expound on what study designs could look like, what objectives those studies should embody, and what interventions can best be used or adapted in the near future for control a shifting epidemiological situation.

**Editorial and Data Presentation Modifications?**

Reviewer #1: See comments about Results and general comments to the authors.

Reviewer #2: In Figure 4 in the legend there is a misspelled word: Open circles are data points more than 24 months after the intervention and where excluded for the analysis.

Meaning of sentence in Line 780 was not clear. There was no evidence of a change in clinical case attributed to P. vivax over time was estimated for IRS.

**Summary and General Comments**

Reviewer #1: The manuscript by Melanie Loeffel & Amanda Ross describes a thorough review of the available literature on the impact of three major interventions (treated bednets, ITN; indoor residual spraying, IRS; mass drug administration, MDA) in areas of sympatric vivax and falciparum malaria. The ability to compare dozens of studies side by side as the authors present is of great value to scientists in the field and policy makers when considering future control initiatives.

The approach and methods are clearly described in detail, but the presentation of results is a bit tedious and overwhelming, particularly when describing figures 3 - 20. To avoid distractions and sole description of the graphs, the authors may benefit of rearranging the presentation and transferring these figures to supplemental material. It will require a little creativity to still briefly describe those results in the main text, while highlighting the overall “big picture”, which is shown in figure 21. All the other tables and boxes from the main text are important are clearly presented and should be kept as they are. The interesting finding of the analysis is the great variability observed on vivax malaria prevalence post-intervention. However, such variability highlights caution against over interpretation of the data from figure 21. Finally, the discussion section can be improved to reduce repetition of statements used in the results and to highlight the importance to take into consideration the biological differences between vivax and falciparum when evaluating a general intervention not targeting hypnozoites. Below are some specific points for consideration:

1. Page 4, line 104: please explain the discrepancy between “973 hits” and “1157” hits described in page 14, line 361.

2. Page 5, line 112: did the authors find lots of reports in Chinese or Portuguese that had to be discarded due to language? These are regions where Pv and Pf occur and many studies are published only in magazines available to their countries and their languages.

3. Page 11, lines 281-285: please spell out or quote the definition of relapse patterns described in reference 27. The cited review article has an enormous amount of information and analysis of relapses, so it will be helpful to read the definition the authors decided to use. For example, figure 22 of reference 27 highlights the possibility of confusion caused by similar relapse patterns with long-latency and frequent relapsing parasites. 

4. Page 12, line 303: to help the readers the authors should write the minimum sample size of the studies and refer to which supplemental material shows the extracted data.

5. Page 23, Figure 3: the figure font is too small, but if the authors transfer the graphs to supplemental data they can be presented in full size. To facilitate visualization of the different regions, the titles could be shorter to include just country and city/region, and a background color coded by country. That would quickly show the patterns observed in Cambodia (Fig3 n- v), where primaquine is rarely provided due to high prevalence of G6PD deficiency. This could explain the steady prevalence of vivax malaria and is supported by the fact that most vivax cases following chloroquine treatment in Cambodia are associated with hypnozoite reactivation (Popovici J et al, JID 2019). 

6. Page 24, Figure 4: graph titles are too long, therefore cut in some cases. Please check if the axis title is correct “time since LLIN distribution in months”

7. Page 25, Figure 5: please clarify if the studies with control groups suggest no difference in the impact of intervention on vivax prevalence.

8. Page 27, MDA: considering the important biological difference between vivax and falciparum regarding development of dormant liver stage in the former, the authors should emphasize that drug interventions focus mostly against blood stages. This is critical as relapses cases will account for greater vivax prevalence post-intervention.

9. Minor text edits should be considered in the lines listed below and a text search and replace should be performed to include a space between “P.” and “vivax”/“P.” and “falciparum”:

• Line 55: spell out Plasmodium vivax instead of P. vivax

• Line 247: missing period after P from P vivax.

• Line 289: space after seasonality and before comma

• Line 529: “…were had received…”

• Line 540: “…to rid of…”

• Line 598: “…there it was likely that…”

• Line 765: “signifiicant”

Reviewer #2: (No Response)

PLOS authors have the option to publish the peer review history of their article (what does this mean?). If published, this will include your full peer review and any attached files.

Reviewer #1: No

Reviewer #2: No
---

## [Decision Letter · Decision Letter 1]

27 May 2022

Dear Dr. Ross,

We are pleased to inform you that your manuscript 'The relative impact of interventions on sympatric Plasmodium vivax and Plasmodium falciparum malaria: a systematic review' has been provisionally accepted for publication in PLOS Neglected Tropical Diseases.

Best regards,

Rhoel Ramos Dinglasan

Associate Editor

Marcelo Ferreira

Deputy Editor

Reviewer's Responses to Questions

**Key Review Criteria Required for Acceptance?**

**Methods**

-Are the objectives of the study clearly articulated with a clear testable hypothesis stated?

-Is the study design appropriate to address the stated objectives?

-Is the population clearly described and appropriate for the hypothesis being tested?

-Is the sample size sufficient to ensure adequate power to address the hypothesis being tested?

-Were correct statistical analysis used to support conclusions?

-Are there concerns about ethical or regulatory requirements being met?

Reviewer #2: (No Response)

**Results**

-Does the analysis presented match the analysis plan?

-Are the results clearly and completely presented?

-Are the figures (Tables, Images) of sufficient quality for clarity?

Reviewer #2: (No Response)

**Conclusions**

-Are the conclusions supported by the data presented?

-Are the limitations of analysis clearly described?

-Do the authors discuss how these data can be helpful to advance our understanding of the topic under study?

-Is public health relevance addressed?

Reviewer #2: (No Response)

**Editorial and Data Presentation Modifications?**

Reviewer #2: (No Response)

**Summary and General Comments**

Reviewer #2: (No Response)

PLOS authors have the option to publish the peer review history of their article (what does this mean?). If published, this will include your full peer review and any attached files.

Reviewer #2: No

---

## [Editor Report · Acceptance letter]

14 Jun 2022

Dear Dr. Ross,

We are delighted to inform you that your manuscript, "The relative impact of interventions on sympatric *Plasmodium vivax* and *Plasmodium falciparum* malaria: a systematic review," has been formally accepted for publication in PLOS Neglected Tropical Diseases.

Best regards,

Shaden Kamhawi

co-Editor-in-Chief

Paul Brindley

co-Editor-in-Chief
